# Variability conceals emerging trend in 100yr projections of UK local hourly rainfall extremes

Elizabeth J. Kendon [1,2] ✉, Erich M. Fischer [3] & Chris J. Short [1]

Extreme precipitation is projected to intensify with warming, but how this will manifest locally through time is uncertain. Here, we exploit an ensemble of convection-permitting transient simulations to examine the emerging signal in local hourly rainfall extremes over 100-years. We show rainfall events in the UK exceeding 20 mm/h that can cause flash floods are 4-times as frequent by 2070s under high emissions; in contrast, a coarser resolution regional model shows only a 2.6x increase. With every degree of regional warming, the intensity of extreme downpours increases by 5-15%. Regional records of local hourly rainfall occur 40% more often than in the absence of warming. However, these changes are not realised as a smooth trend. Instead, as a result of internal variability, extreme years with record-breaking events may be followed by multiple decades with no new local rainfall records. The tendency for extreme years to cluster poses key challenges for communities trying to adapt.

Recent floods across Europe have reinforced the need for a better understanding of how rainfall will change locally over the coming decades with global warming. Serious flooding in central Europe in July 2021 resulted in more than 200 fatalities and considerable damage to infrastructure, with estimated costs for Germany alone of €4.5–5.5 billion. This event has been linked to climate change, with climate change increasing the intensity of daily rainfall by 3–19%[1], raising public awareness of the increasing risks from surface water flooding. Although modelling studies agree on a future increase in the occurrence and intensity of extreme rainfall events on global to regional scales[2–6], how this will manifest locally and through the coming years and decades is far from clear. Of importance is when changes in rainfall extremes are expected to 'emerge', i.e., move outside of what has been experienced in the past due to natural climate variability[7]. Recent trends in observed local rainfall extremes have been reported[8–10], with some evidence of these already being influenced by climate change[11]. However, formal detection and attribution at the local scale are not yet possible, with the detection of changes in local hourly extremes expected from the 2040s[12]. An understanding of these changes is vital to support policy decisions encompassing, for example, land management, infrastructure design, and investment in flood defences. In England alone, an estimated three million properties are at risk from surface water flooding[13].

In this study, we are moving beyond projections of rainfall extremes for a short future period to how changes are actually realised through time. We also go beyond previous studies looking at the emergence of trends in daily rainfall extremes[7], to focus on extreme hourly precipitation typically responsible for flash flooding. This type of flooding where water levels rise quickly is particularly destructive, with small steep catchments and urban areas especially vulnerable[14]. Recent events in the UK include flash flooding in Birmingham in May 2018 that led to a fatality[13], and London in July 2021.

Previous studies looking at changes in local rainfall extremes have either used models that are too coarse resolution to reliably capture sub-daily extreme precipitation[15], or are limited to single realisations[16,17] or relatively short future periods[18,19] due to computational cost. A recent advance is ensembles of climate simulations at kilometre scale[20–23]. These 'convection-permitting' models (CPMs) explicitly represent convection providing reliable projections of sub-daily precipitation[24,25]. The UK Climate Projections (UKCP) project provided the first national climate scenarios at convection-permitting scale (UKCP Local[20]), consisting of an ensemble of projections over the

[1]Met Office Hadley Centre, Exeter, UK. [2]Faculty of Science, Bristol University, Bristol, UK. [3]Institute for Atmospheric and Climate Science, ETH Zurich, Zurich, Switzerland. ✉e-mail: elizabeth.kendon@metoffice.gov.uk

UK downscaling different versions of the Met Office Hadley Centre model. Coordinated multi-model ensemble CPM experiments have also been carried out over Central Europe[21,23], identifying the extent to which changes in local rainfall extremes are robust across different climate models. However, given the high computational cost, these state-of-the-art CPM ensemble projections all consist of relatively short time-slice simulations. Aggregation or regional pooling techniques[26] have been used to help identify changes in extremes, for such relatively short climate change realisations[20,27]. However, these techniques assume that rainfall has the same characteristics across the entire region and cannot robustly represent changes in local rainfall extremes.

Here we introduce the first ensemble of long (100-year) convection-permitting climate projections, unique experiments allowing us to overcome these limitations. The ensemble consists of 12 members at 2.2 km resolution over the UK providing data hour by hour, for every 2.2 km grid box, from 1981–2080. The projections represent 12 equally plausible realisations of the evolution of the climate assuming continued increases in greenhouse gas emissions under a high RCP8.5 emissions scenario. This is like starting twelve weather forecasts and running for 100 years, except that we are not interested in the detail on a given day but rather the statistics of extremes year-by-year. These transient CPM simulations allow us to more robustly quantify the local scaling of hourly extreme precipitation[27] and the behaviour of record-breaking events.

The need to include the effects of year-to-year variability alongside projections for a typical average season has been previously shown[28], but only for coarse-resolution climate models. Previous work showed how individual cold winters may continue despite the expectation of the UK having 'hotter drier summers and warmer wetter winters' in future, and that these messages are not contradictory. By considering year-to-year variability in the projections, the scope for misinterpreting observed individual extreme seasons in the context of climate change is reduced. Here we carry out a similar analysis, but for local hourly precipitation extremes, which display greater variability and have more of an impact for some sectors of society.

## Results

### Model performance in capturing extreme hourly precipitation

We assess the performance of the CPM compared to the driving 12 km regional climate model (RCM) in simulating hourly precipitation. Both ensembles consist of twelve members for 1981–2080 under RCP8.5, with the CPM spanning the UK and the RCM spanning Europe. The CPM has similar model physics to the RCM, but at higher (2.2 km) resolution it is possible to switch off the convection-parameterisation scheme resulting in an explicit representation of convection. We show that the CPM gives a much better representation of the number and intensity of hourly rainfall extremes, whereas the representation is seriously biased even at the comparatively high resolution of 12 km in the RCM. The simulated annual maximum hourly precipitation in the CPM agrees well with hourly gridded observations (CEHGEAR; a gridded hourly rainfall dataset spanning Great Britain based on hourly and disaggregated daily rain gauges, see Methods) from 1991–2014 (shown for South-East and North-West England in Fig. 1). By comparison, the 12 km RCM strongly overestimates the observed hourly values with regional maxima being much higher and more variable than in the CPM and observations. This regional maximum corresponds to the maximum local hourly precipitation simulated within the region (considering all 12 km grid boxes and all hours) in a given year. The observed multi-year mean and year-to-year variability of regional maximum values is within the 5-95% range for the CPM, but outside the range for the RCM, an indication that the bias is systematic (Fig. 1d). A similar result is found if we consider seasonal instead of annual maximum values (Supplementary Figures S1–S2). The findings are consistent across other regions in England and Wales (Supplementary

Figure S3), with the CPM showing good agreement with the observations in terms of the regional maximum values and their year-to-year variability. Over Scotland, the CPM underestimates regional maximum values, with the RCM giving apparent better agreement. The tendency for the RCM to overestimate regional maxima is less apparent for local maxima, where annual maximum values locally can be considerably less than the regional maxima (Fig. 2). However, the RCM consistently overestimates year-to-year variability in local annual maximum values, with the CPM giving better agreement, except over northern Scotland. Over northern Scotland, CEHGEAR suggests much higher values of local annual maximum hourly precipitation over high ground in the north-west compared to both models. However, we have less confidence in the observational values in this region due to a lack of availability of hourly gauges, leading to hourly values being estimated by disaggregating daily gauge totals using an average storm profile[29]. Although in general, we expect an underestimation of heavy rainfall due to gauge under-sampling and systematic under-catch (see Methods), particularly over regions of high elevation[30], the disaggregation step may lead to peak hourly values being overestimated in CEHGEAR in regions of persistent orographic rainfall.

The frequency of hourly events exceeding a high threshold is also better captured in the CPM than the RCM. Figure 1c shows the frequency of events exceeding 20 mm/h, for rainfall averaged over a 12 km grid box, corresponding to events that can potentially produce serious damage through flash floods (see Methods). The RCM again significantly overestimates the number of events across the UK, with the observed value outside the 5–95% ensemble range (Fig. 1d). The CPM shows good agreement with the observations for exceedance frequencies of other thresholds examined too (Supplementary Figure S4), whilst the RCM underestimates the number of events exceeding 5 mm/h and overestimates the number of events exceeding 20 mm/h and 30 mm/h.

The much better representation of hourly precipitation in the CPM compared to the RCM is also seen seasonally. The UK-averaged biases in seasonal mean and heavy hourly precipitation are consistently lower in the CPM than RCM and even smaller than observational uncertainty (Supplementary Figures. S5–6). These results support the credibility of the CPM in providing reliable present-day representation and future projections of hourly precipitation extremes. The tendency for the RCM to underestimate the occurrence of moderate extremes (5 mm/h threshold, corresponding approximately to the 99.9th percentile of hourly precipitation) is consistent with previous studies that have shown the tendency for regional climate models to underestimate heavy hourly rainfall[31]. This underestimation has been linked to deficiencies in the convection-parameterisation scheme[32,33], which have been designed to capture the average effects of convection and not individual extreme events. The overestimation of very rare precipitation extremes (20 mm/h and 30 mm/h thresholds) in the RCM is likely a reflection of unphysical grid-point storms in the model where the assumptions underlying the convection-parameterisation scheme break down. Such an event is not physical but instead is a numerical instability in the model, and arises due to violation of the assumption that convective clouds have areas much smaller than the grid box. This leads to unphysically high values of >100 mm/h at the 12 km scale (Fig. 1, Supplementary Figure S3), which are considerably higher than observed maximum values even accounting for gauge under-catch (see Methods), and gives us low confidence in future hourly extremes from the RCM[25].

For the remainder of this paper, we focus on the analysis of future changes in hourly precipitation extremes in the CPM. However, differences with respect to the RCM (shown in Supplementary Material) are discussed to illustrate how the novel CPM experiments change our understanding over existing model evidence.

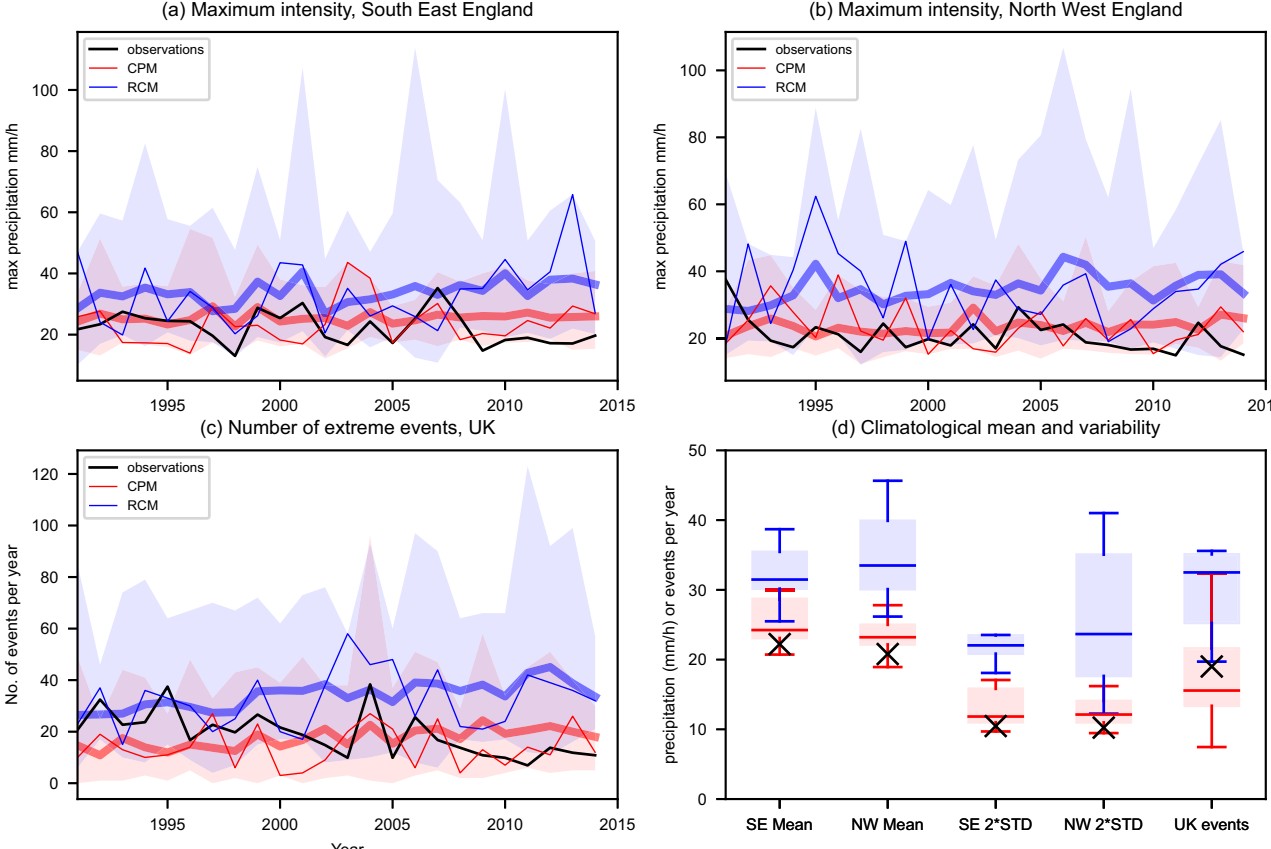

**Fig. 1 | Model performance in representing extreme hourly precipitation.**
Regional annual maximum hourly precipitation for **a** South-East England and **b** North-West England, and **c** number of events per year across the UK exceeding 20 mm/h. Regional annual maximum corresponds to the maximum precipitation (mm/h) occurring within the region (considering all 12 km grid boxes) in a given year. Results are shown for CEHGEAR observations (black), the 2.2 km convection-permitting model (CPM, red) and 12 km regional climate model (RCM, blue) for the standard member (thin line), ensemble-mean (thick line) and ensemble min–max range (shaded) for 1991–2014, for hourly rainfall averaged over 12 km grid box. The standard member is illustrative of the variability in an individual realisation. Also shown **d** is a box plot of the 24-year mean and standard deviation (scaled by 2, 2*STD) of regional annual maximum values (for SE and NW England) across the ensemble (measured as precipitation mm/h), and similarly for mean events per year across the UK (measured as events per year). The red (blue) horizontal line shows the median for the CPM (RCM), shaded box the 25–75th percentile range, and the whiskers the 5th–95th percentile range; observed values are shown as a black cross.

## Future changes in extreme hourly precipitation

The temporal evolution of regional annual maximum hourly precipitation, for SE England and NW England, from 1981 to 2080 in the CPM is shown in Fig. 3 (and for the RCM in Supplementary Figure S7). This corresponds to the maximum local hourly precipitation simulated within the region (considering all 12 km grid boxes and all hours) in a given year. Visual inspection of the ensemble-mean shows that there is an underlying increasing trend, which is seen for all regions and seasons (Supplementary Figures S1–S3). However, there is considerable variability from year-to-year and this variability generally increases with time (Fig. 3c). This increase in interannual variability is largely in proportion to the mean, reflecting the increase in extreme hourly precipitation through time. However, there is some evidence that interannual variability in extremes is higher for some periods than others, e.g., 2040s and 2050s over SE England (Fig. 3c). It is likely such features relate to multi-decadal natural variability. We note that the UKCP18 global simulations display a strong weakening of the Atlantic Meridional Overturning Circulation (AMOC), which translates into increased storminess[34], and also in the 2040s the Arctic becomes seasonally ice-free under RCP8.5 and there are possible links between Arctic amplification and mid-latitude severe weather[35]. However, further work is needed to identify if any such mechanisms lead to increased interannual variability common to all ensemble members. In general, the considerable interannual variability in precipitation extremes makes it difficult to discern any underlying trend in an individual realisation[12]. Thus, the use of a large ensemble is vital to isolate the underlying climate change response. Our unique ensemble of convection-permitting simulations allows us to robustly quantify changes in the intensity and frequency of hourly precipitation extremes in subregions of the UK.

We examine the number of events per year across the UK exceeding 20 mm/h (Fig. 3d). This allows us to assess how the interplay between the underlying trend in extreme precipitation and year-to-year variability manifests in terms of the number of events exceeding a specific threshold, important from an impacts perspective. In the CPM, there is a large 4x increase in the number of events exceeding 20 mm/h, increasing from 12 per year in 1980s to 49 per year in 2070s for the UK as a whole, under high emissions (Fig. 4). In the RCM, events are only 2.6× as frequent (Supplementary Table S1). Thus, the RCM not only overestimates the occurrence of extreme hourly events in the present-day climate, it also likely underestimates their relative future changes. A similar result is found for a lower threshold of 10 mm/h, with the RCM showing smaller increases in the number of events than the CPM, although the differences between the CPM and RCM are less marked for the lower threshold (Supplementary Table S1). In general, the higher the threshold the greater the percentage of future change in the number of events.

Regionally, future changes in the number of events exceeding 20 mm/h varies, with the largest increases in the north of the UK. For example, in North Scotland events are almost 10× as frequent in the

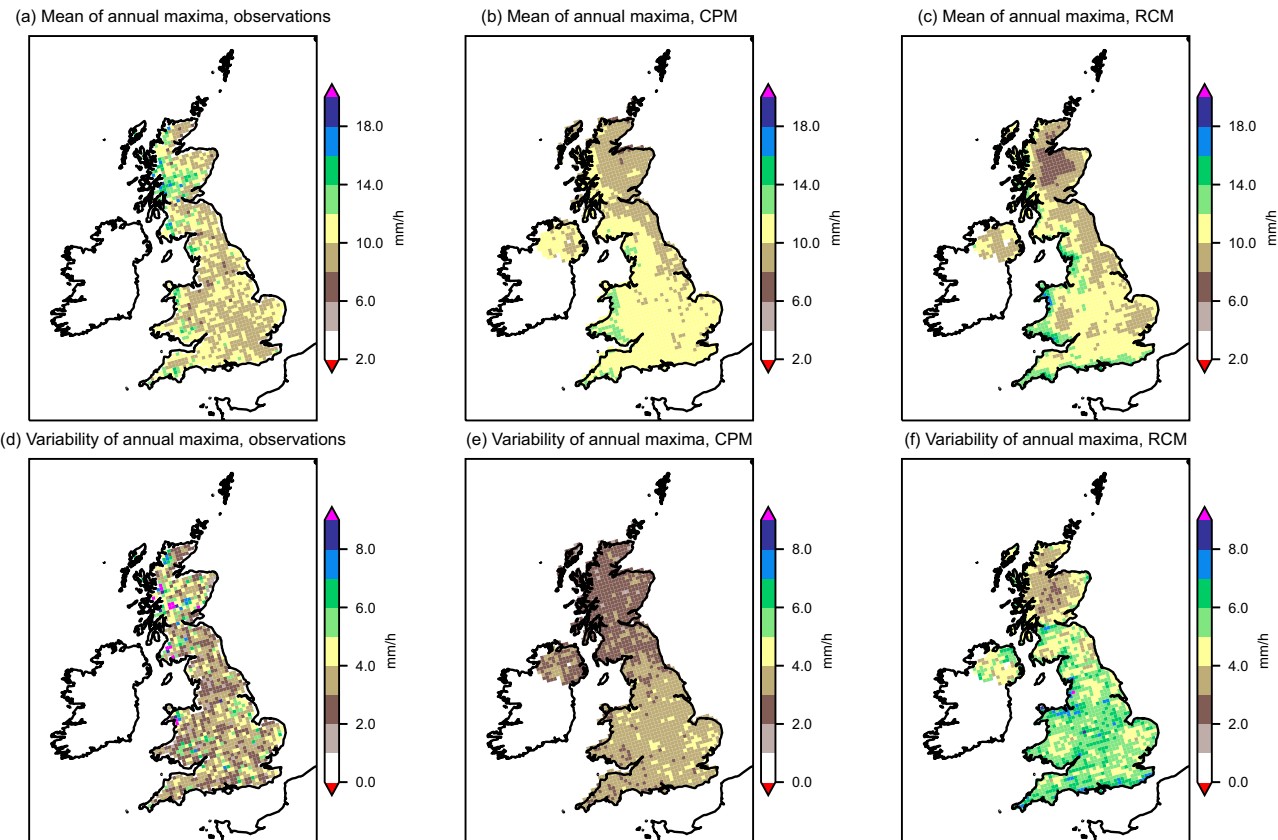

**Fig. 2 | Model performance in representing local maximum hourly precipitation intensity.** Present-day **a**–**c** mean and **d**–**f** standard deviation of annual maximum hourly precipitation at every 12 km grid box, for 1991 to 2014. Results are shown for **a**, **d** CEHGEAR observations and the median of the **b**, **e** 2.2 km convection-permitting model (CPM) and **c**, **f** 12 km regional climate model (RCM) ensembles.

2070s compared to the 1980s in the CPM (Fig. 4). In the south of the UK, the value is typically ~3×. At the grid-box scale, there are few events (much less than 1 event every 10 years) exceeding 20 mm/h in the 1980s. By the 2070s, in the CPM, such events are more common everywhere, with an event on average every 5–10 years over many grid points in the west (Supplementary Figure S8). Locally events can be >10× as frequent, especially over parts of Scotland. However, the exact magnitude of these changes is not robust given the few events in the present day. Regionally, future relative changes are consistently lower in the RCM compared to the CPM (Supplementary Table S1). On using a percentile threshold (99.99th percentile of present-day hourly precipitation, corresponding to 9.28 mm/h in the CPM and 9.35 mm/h in the RCM), these differences are reduced, however for almost all sub-regions, increases are still lower in the RCM than CPM (Supplementary Table S1).

While the CPM ensemble allows us to reliably quantify the long-term signal in hourly precipitation extremes, this increase is not realised as a smooth trend. Looking at the individual ensemble member realisations, there is considerable variability from one year to the next (Fig. 3d; RCM in Supplementary Figure S4). This variability from year-to-year increases with warming, which makes adaptation particularly challenging. Extreme years (with lots of extreme rainfall events) can occur by chance even in the present climate, whilst years with few events can still occur late into the 21st century. For example, in one CPM realisation there is a year in the 1980s with >70 events across the UK exceeding 20 mm/h; equally there are years in the 2070s with only 10 events. However, there is clearly an increase in the statistically expected likelihood of events exceeding 20 mm/h through time and years with less than five events are entirely absent by the 2070s.

For a smaller region, such as London, the underlying trend in the number of events is harder to discern and is not evident from a single realisation (Supplementary Figure S9).

## Temperature scaling of extreme precipitation variability and change

We now explore the relationship (termed scaling) between extreme precipitation and temperature. This allows us to translate changes through time for high emissions to changes per degree of warming, applicable to lower emissions scenarios or specific global warming levels. It is also important for process understanding. The scaling relationship allows us to assess whether increasing moisture with warming is dominating changes on year-to-year and/or multi-decadal timescales. In particular, the moisture holding capacity of the atmosphere increases at ~7% per degree Kelvin temperature rise, the 'Clausius-Clapeyron' (CC) relationship, and this has often been used to set the scale for changes in precipitation extremes[36].

The scaling of regional annual maximum precipitation with annual mean temperature, for two UK regions, is shown in Fig. 5 (panels a and c). For the smoothed climate change signal (panels b and d, see Methods), scaling coefficients (expressed as a percentage increase in precipitation per degree temperature rise) agree well with CC-scaling of 7%/K. Results for other regions are shown in Fig. 6, with scaling coefficients for annual extremes ranging between 5.2–14.8%/K (Supplementary Table S2). In general, values are close to CC-scaling for southern/central regions, with higher values for northern regions. Local scaling coefficients at the grid box scale (Fig. 7) show a similar north-south gradient, with values close to or below CC scaling in the south and east. Higher scaling coefficients are seen in the north of the UK and over western coasts, with values above 2xCC locally in the

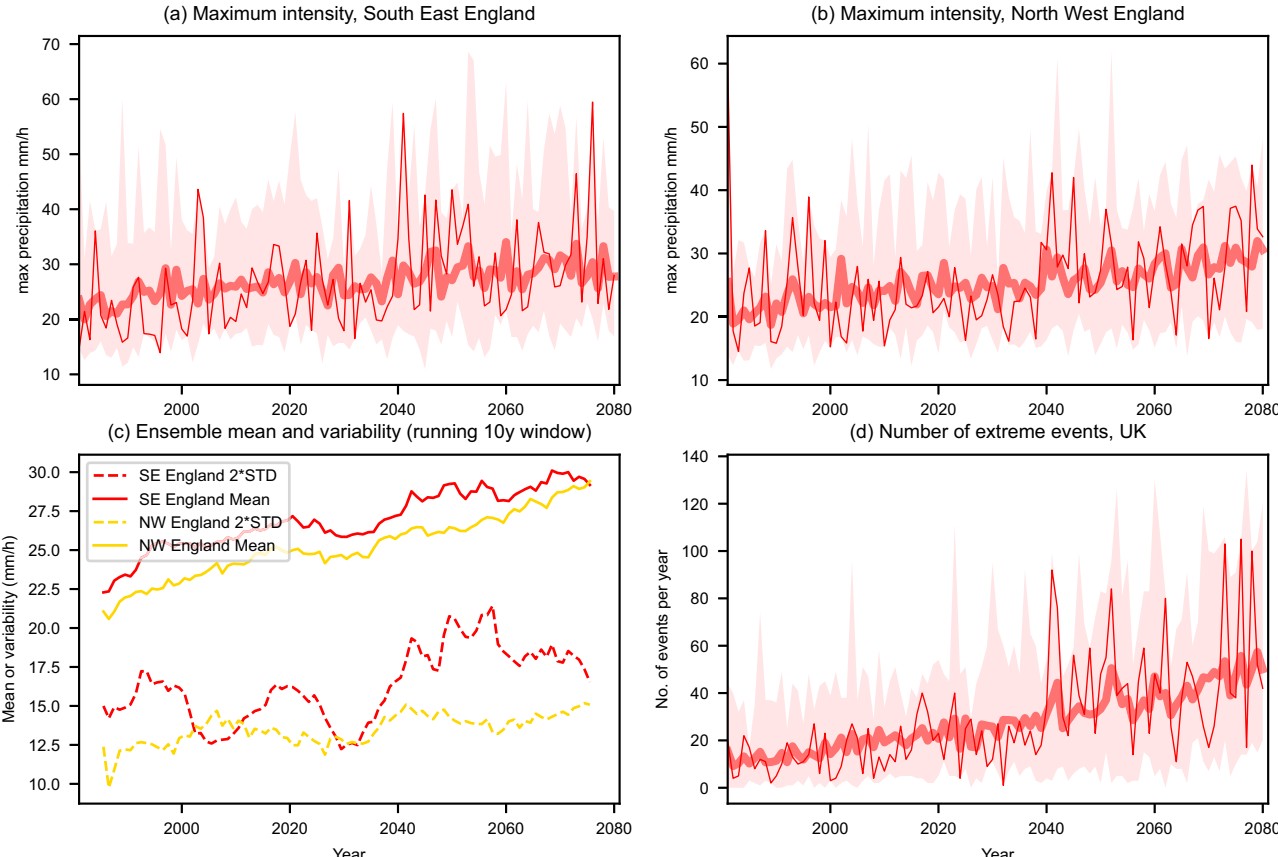

**Fig. 3 | Evolution of hourly precipitation in the convection-permitting model.**
Shown is regional annual maximum hourly precipitation (mm/h) for **a** South-East England and **b** North-West England, for the standard member (thin line), ensemble-mean (thick line) and ensemble min–max range (shaded), for hourly rainfall averaged over 12 km grid box, for 1981–2080. **c** Running 10-year multi-member mean (solid) and variability (standard deviation scaled by 2, 2*STD, dashed) in regional annual maximum values. **d** Number of events per year across the UK exceeding 20 mm/h. Regional annual maximum corresponds to the maximum precipitation (mm/h) occurring within the region (considering all 12 km grid boxes) in a given year. Threshold exceedances occurring within a UK subregion on the same day are considered part of a single event.

north-west. This highlights that the large ensemble allows us to robustly determine the local signal in extreme hourly precipitation. Scaling coefficients are generally higher for winter extremes (range 6.7–13.6%/K) than summer extremes (3.2–13.3%/K), except for northern regions where the reverse is true (Fig. 6).

These results are consistent with increasing atmospheric moisture being the primary driver of the smooth underlying change in the intensity of extreme precipitation, with changes in moisture availability largely explaining the seasonal and regional differences in scaling. In summer, although specific humidity increases, relative humidity decreases[37], implying that moisture availability does not increase with temperature as fast as the temperature-dependent maximum (i.e., following the change in saturated specific humidity). Greater decreases in relative humidity in the south of the UK, also explain the north-south gradient in scaling coefficients. We note that results are very similar on using UK-mean instead of local temperature as the scaling variable (Supplementary Figure. S10), although the tendency for large departures from CC-scaling is reduced. This suggests that some of the low (in the south) and high (in the north) scaling rates (Fig. 7) may be due to moisture availability in weather systems (where the moisture source may be remote) increasing at a different rate than the local temperature-dependent maximum. The appropriate scaling variable will depend on the relative importance of large-scale weather patterns compared to locally triggered convection. However, the similarity of the results suggests that our conclusions here are not sensitive to this choice. In addition to changes in moisture availability,

changes in weather patterns and tropospheric stability may lead to scaling rates that depart substantially from CC[38–41]. Dynamical feedbacks caused by latent heating (and evaporation of rain causing downdrafts and cold pools) have also been cited as a possible explanation for super-CC scaling rates for hourly precipitation extremes[42]. Such local dynamical feedbacks are only captured by CPMs, and thus we may anticipate different scaling rates between the CPM and RCM. For the raw data, scaling coefficients are higher in the CPM than RCM for summer extremes (consistent with the importance of local dynamical feedbacks within convective storms) and typically lower for winter extremes; but the pattern is not straightforward for the underlying climate change signal (Supplementary Table S2). This is due to differences in scaling reflecting many factors, including the presence of grid-point storms in the RCM.

While there is a strong relation to long-term warming, this relationship between extreme precipitation and temperature does not hold on a year-to-year basis. Extreme precipitation intensity shows a lot more scatter for the raw data (Fig. 5, panels a and c), with some members seen to give extreme precipitation intensity considerably away from the regression line (departures are even larger for the RCM, Supplementary Figures S11–12). This shows that annual mean temperature is not a strong constraint for the intensity of extreme precipitation events on a year-to-year timescale (and across ensemble members). This is further highlighted by the relationship for detrended data, where the climate change signal has been removed (Supplementary Figures. S11–12).

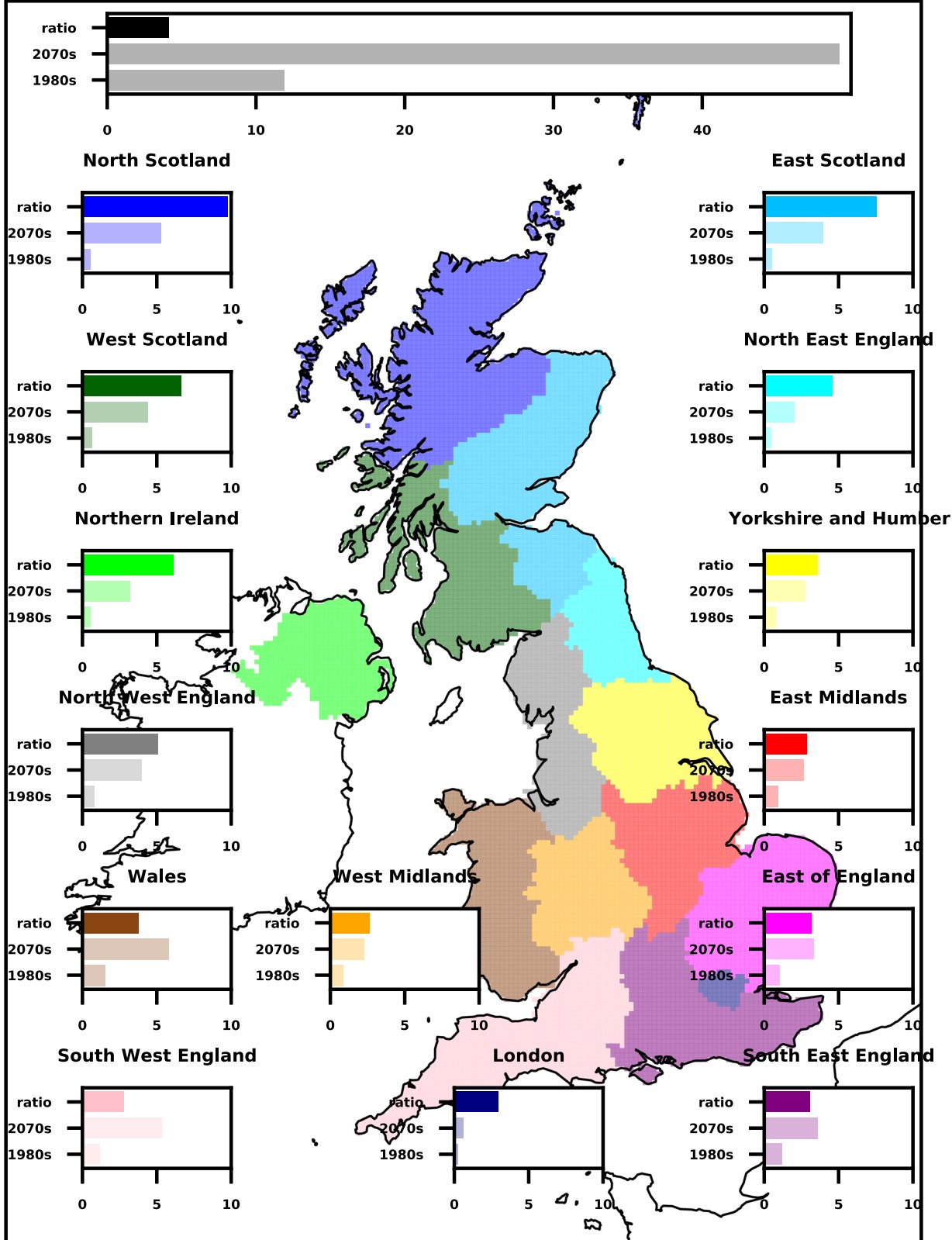

**Fig. 4 | Future changes in the occurrence of extreme hourly precipitation regionally.** Average number of events per year exceeding 20 mm/h in the 1980s and 2070s, and their ratio (future/control), for regions of the UK. Values correspond to the multi-member 10-year mean number of events, for hourly rainfall averaged over 12 km grid box in the convection-permitting model (CPM). Threshold exceedances occurring within a UK subregion on the same day are considered part of a single event. Colours indicate UK subregions, with UK-wide results shown in black at the top.

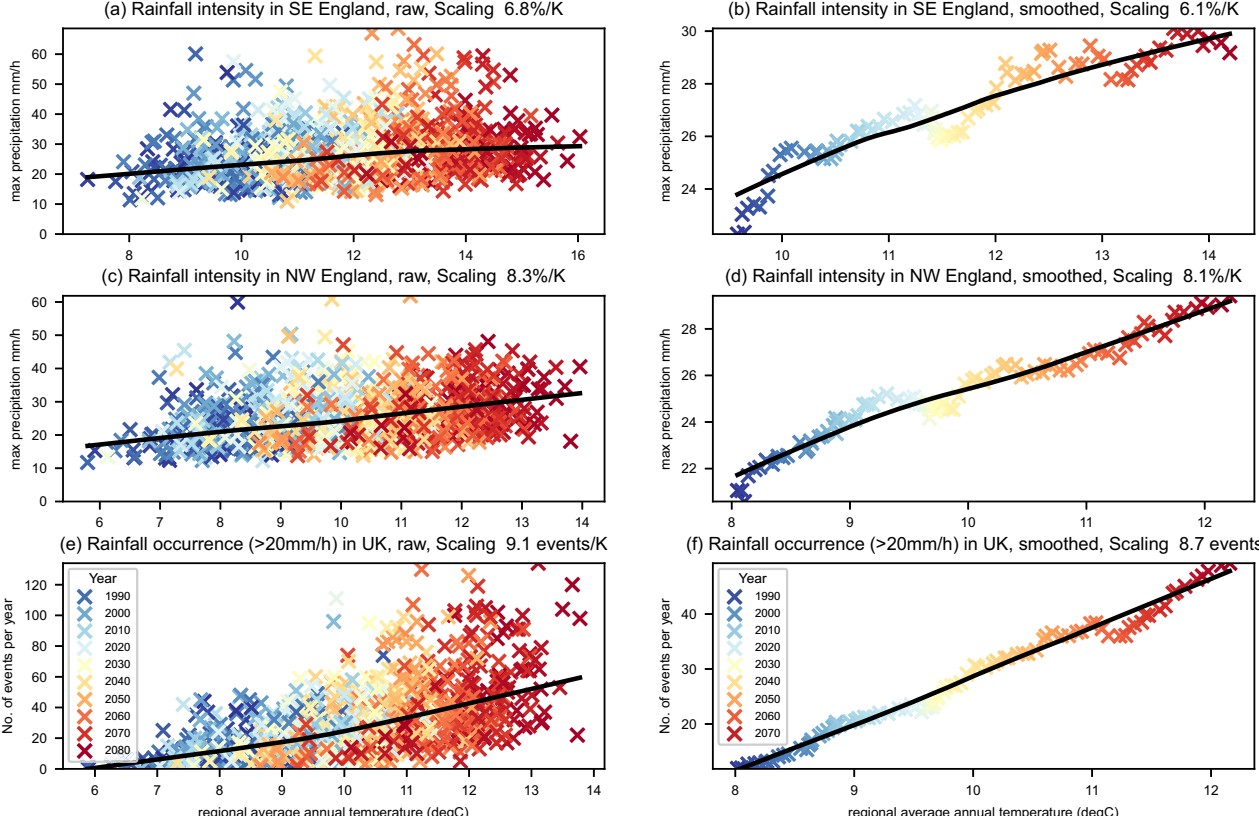

**Fig. 5 | Temperature scaling of intensity and occurrence of extreme hourly precipitation.** The highest annual maximum hourly 12 km precipitation (mm/h) across grid points is plotted against the regional average annual temperature (°C) per year, for **a**, **b** South-East England and **c**, **d** North-West England, in the convection-permitting model (CPM). Also shown is **e**, **f** number of events exceeding 20 mm/h across the UK per year plotted against UK average annual temperature.

For raw data **a**, **c**, **e**, there are 12 points for each year, corresponding to the 12 CPM ensemble members. For smoothed data **b**, **d**, **f**, results are shown for the multi-member 10-year running mean. A lowess regression has been fitted to the data (black line), and colours correspond to the year. Titles give the linear scaling coefficient.

Figure 5e shows the relationship between the number of events exceeding 20 mm/h and annual mean temperature. The smoothed data (Fig. 5f) shows an approximately linear relationship between the occurrence of extreme events and the long-term temperature trend, with an increase of almost 9 events per year per K increase. Again, the relationship for the raw data (or detrended data, Supplementary Figure S13) shows a lot more scatter, with annual mean temperature not a strong constraint for the occurrence of extreme precipitation events on a year-to-year basis. The temperature at the time of extreme precipitation events is a more direct measure of moisture availability, and this will depart considerably from the annual mean temperature due to sub-seasonal variability. In addition dynamical drivers, related to regional circulation patterns, will be important in determining the year-to-year variability in the intensity and occurrence of precipitation extremes.

The scaling coefficient itself is similar between the raw (or detrended) and smoothed data in the CPM for both the intensity and frequency metrics in Fig. 5. This is also true across all regions for annual extremes in the CPM (Supplementary Table S2). For the RCM, large differences in the scaling coefficients between the raw and smoothed data occur, again highlighting issues with the reliability of the RCM scaling results. The similarity in the CPM suggests, to first order, that the increase in the intensity and occurrence of extreme precipitation events with long-term warming can be extrapolated from the present-day relationship providing there are sufficient years of data and/or ensemble members to characterise the average scaling behaviour. However, as shown in Supplementary Figures. S11–13, 25 years of observations are not sufficient, and only a large ensemble as used here

allows us to robustly determine the regional signal in extreme hourly precipitation[4,43]. We also note, for seasonal extremes, extrapolation of the natural variability relationship would lead to an under (over) estimation of the influence of climate change on the intensity of winter (summer) extremes (Supplementary Table S2).

Overall, these results suggest that the complex interplay between natural variability and the underlying trend in the occurrence and intensity of extreme precipitation through time, realised differently in the different ensemble members, cannot simply be understood from the temporal evolution of temperature.

## Exceedance of records through time

The exceedance of records is something that captures the wider public interest, and often raises the question of the role of climate change. Here we exploit our unique CPM ensemble to examine the extent to which climate change has an imprint on the occurrence of hourly precipitation records through time. This directly relates to the question of when a climate change signal in local rainfall extremes is emerging from variability.

We consider annual (or seasonal) maximum hourly precipitation, taking the maximum value sampled at any grid box across the UK (or individual regions). Thus, we do not allow multiple records within a given year or season. Instead, we focus on how often on a year-by-year basis records are exceeded and by what margin, and to what extent this is impacted by climate change. It is important to note that from theory the frequency of records is expected to decrease the longer we measure[44,45], as is the margin by which they are broken[46].

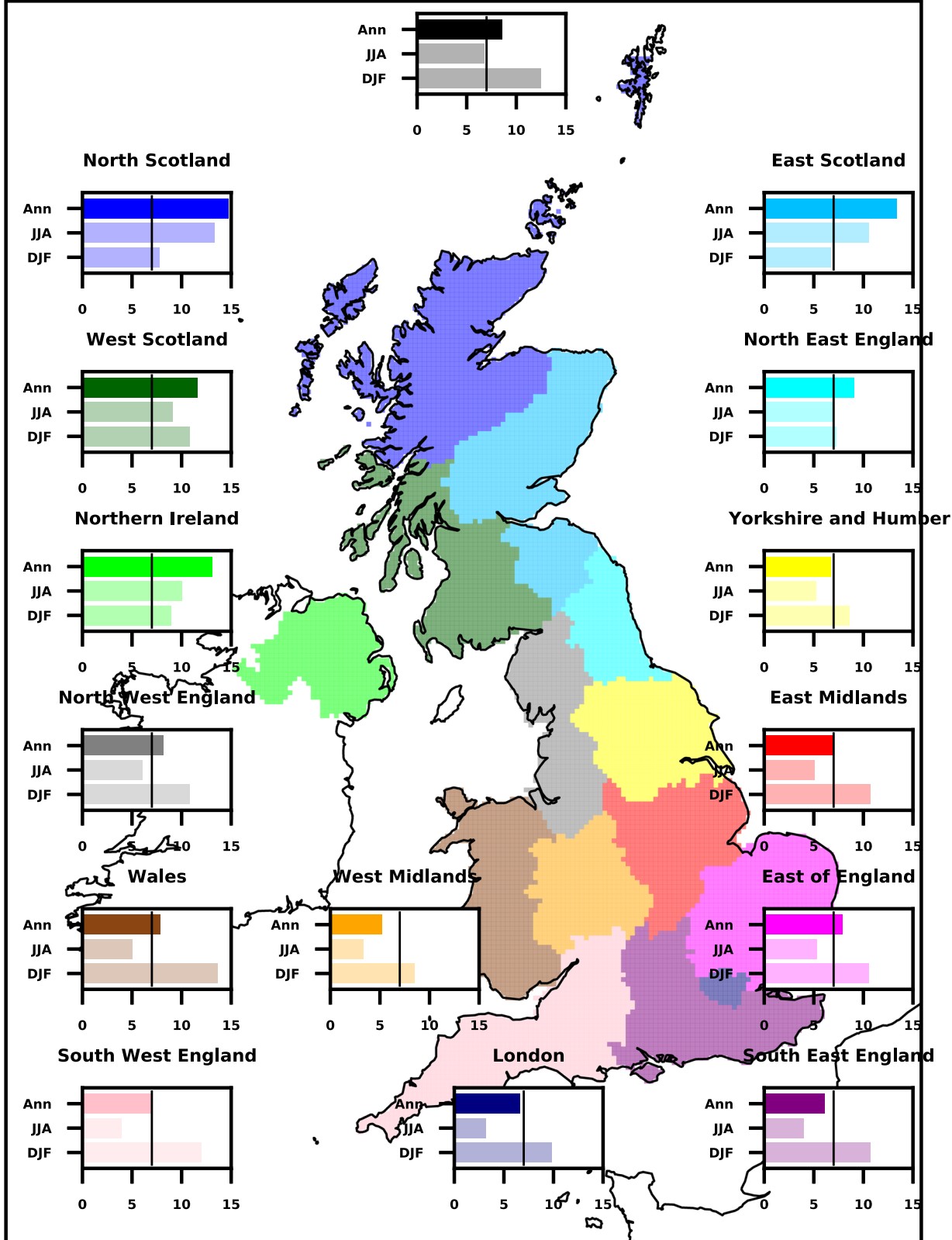

**Fig. 6 | Temperature scaling of underlying change in regional maximum hourly precipitation.** Scaling coefficients (%/K) for underlying climate change signal in regional maximum hourly precipitation in the convection-permitting model, for regions of the UK. Values correspond to the gradient of a linear trend line fitted to smoothed yearly (annual, Ann) [or seasonal (December–January–February DJF or June–July–August JJA)] maximum precipitation versus annual [or seasonal] mean temperature, expressed as % increase in precipitation per K temperature increase. The smoothed data consists of the multi-member 10-year running mean. Colours indicate UK subregions, with UK-wide results shown in black at the top. The black vertical line indicates the Clausius-Clapeyron scaling of 7%/K.

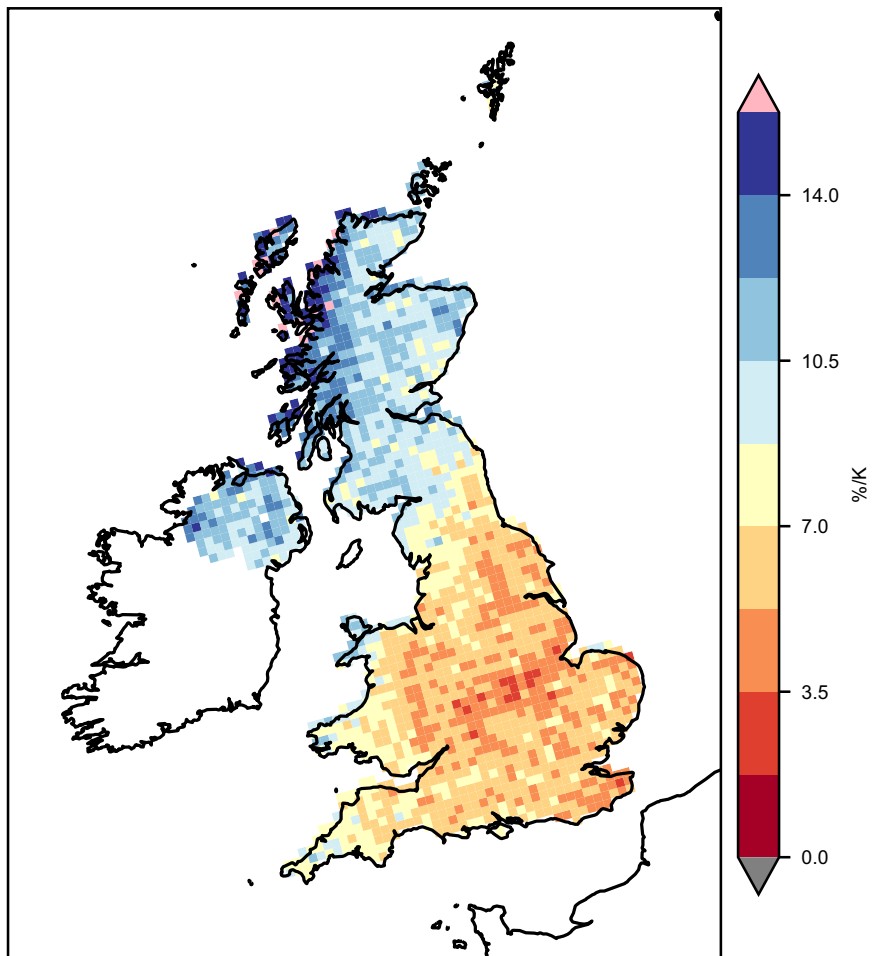

**Fig. 7 | Temperature scaling of underlying change in local maximum hourly precipitation.** Scaling coefficients (%/K) for underlying climate change signal in local annual maximum hourly precipitation in the convection-permitting model (CPM). Values correspond to the gradient of a linear trend line fitted to smoothed annual maximum precipitation versus local annual mean temperature at the 12 km scale, expressed as % increase in precipitation per K temperature increase. The smoothed data consists of the multi-member 10-year running mean.

Figure 8 shows the exceedance of UK records of local hourly rainfall as a function of year from the start of the dataseries. For the observations we only have 25 years of data, whilst for the CPM we have a 100-year timeseries for each ensemble member. As expected, the average number of records per year decreases with time from the start of the dataseries (Fig. 8c) but slower than in the absence of climate change. Figure 8d compares the ensemble-mean rate of decay in the raw data to that for the detrended data, which shows the exceedance of records through time solely due to natural variability (Supplementary Figure S14). The number of records is consistently higher in the raw data, with the ratio of raw to detrended counts ~1.2. This suggests that the exceedance of records in the raw data is dominated by natural variability, but for any given year the probability of experiencing a record is ~20% higher than in the absence of climate change. This record probability is derived for 100 years of data and 12 ensemble members, whereas for an individual realisation, it is not possible to discern an imprint of climate change on the occurrence of records for 100 years of data. If we consider the exceedance of regional records accumulated across the UK, then the ratio of raw to detrended counts becomes 1.4, showing a stronger influence of climate change (Supplementary Figure S15). UK-wide records are of national importance and often reported in the media, whilst regional records may be of more interest to most stakeholders, since they correspond to any region being hit by a locally unprecedented event.

Figure 8b shows when individual records are set and the extent to which they represent an increase on the previous value. In the CPM, large record jumps (>10 mm/h) diminish in number with time, but can occur even after 80 years. On removing the long-term trend in the data, large jumps remain (Supplementary Figure S14), showing that natural variability is the primary driver of records being broken by a considerable margin. We note very large record jumps of >60 mm/h in the RCM, leading to record values of >100 mm/h (Supplementary Figure S14), are not realistic and likely reflect grid-point storms.

Despite the underlying trend in the data, there can be long periods of time in a given realisation when no new records are set. For example, the standard member records 63mm/h in the early 1990s (after 12 years of data) but then no new records for another 70 years until a new value of ~74mm/h in the 2060s (after 83 years of data, Fig. 8a). Thus, a lack of record rainfall values, even after many decades of observations, is not evidence of an absence of an underlying increasing trend in extreme rainfall. The occurrence of records, certainly for rainfall on small space and timescales (here 12 km and hourly) is dominated by natural variability. Only with more data (i.e., a long multi-centennial timeseries or in this case use of multiple model realisations) can the impact of climate change on the occurrence of records be detected. Although this indicates an emergence time of well beyond this century for changes in the occurrence of records, changes in a less extreme measure of local hourly rainfall are expected to emerge much sooner (and within the next few decades[12]).

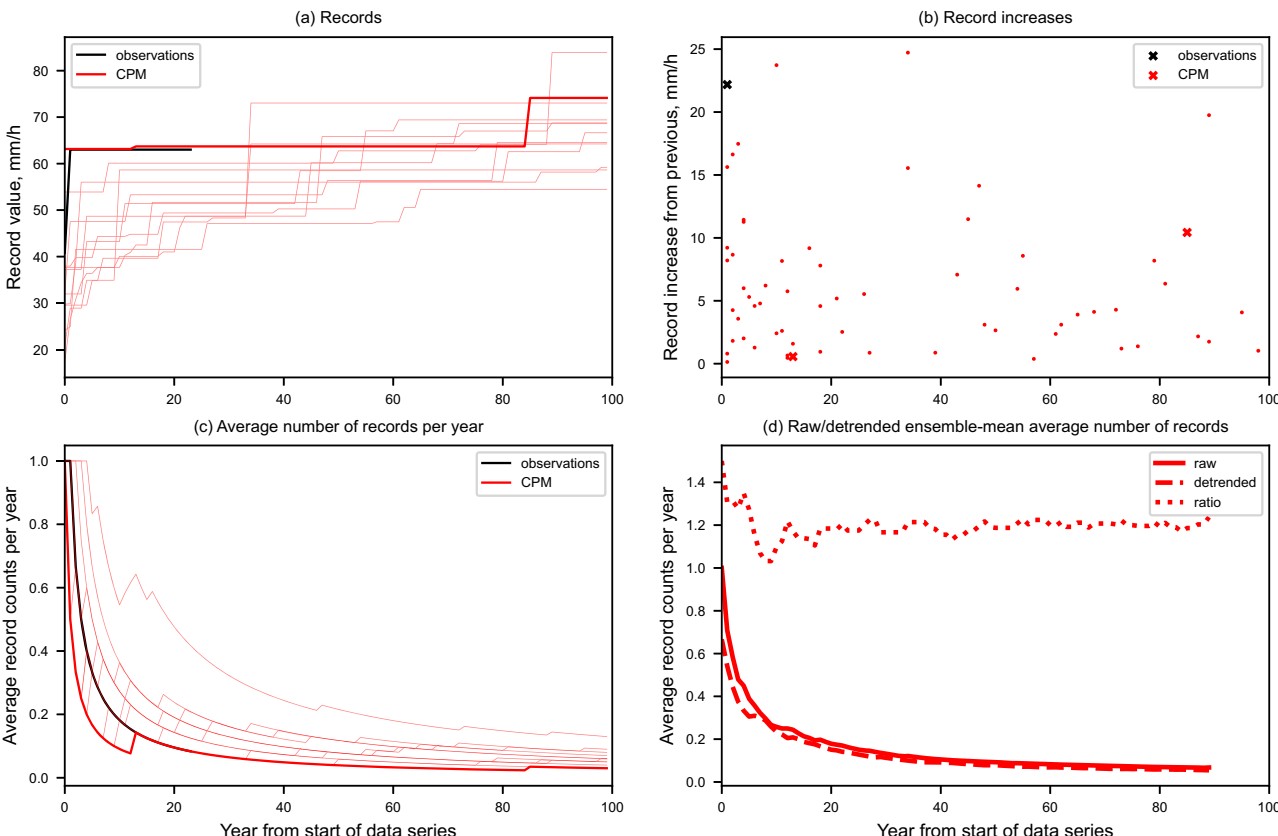

**Fig. 8 | Exceedance of UK records in local hourly precipitation.** Exceedance of records as a function of year from the start of the dataseries. **a** Highest value (mm/h) of annual maximum hourly precipitation recorded at any 12 km grid-point across the UK since the start of the dataseries, **b** increase in record from the previous values set, and **c** average number of records per year from the start of the dataseries. Results are shown for CEHGEAR observations (black) for 1991–2014 and the convection-permitting model (CPM) ensemble members (red) for 1981–2080, with the standard member shown as bold (or red cross) and other members as thin lines (or red dot). Also shown **d** is the CPM ensemble-mean average number of records per year for the raw data (thick solid), detrended data (thick dashed) and their ratio (thick dotted). Detrended data is where the multi-member 10-year running mean of annual maximum values has been removed.

Similar results are found for local hourly rainfall extremes in a given season (Supplementary Figures S16, 17). The standard member shows a gap of 90 years between UK-wide winter records and 70 years for summer records. However, for the ensemble mean, climate change is found to increase the occurrence of records by 30% in winter and 25% in summer. For records across a smaller region (e.g., London, Supplementary Figure S18), larger record jumps are possible, e.g., almost 40 mm/h in one CPM realisation after 80 years.

Although the occurrence of records for local hourly rainfall is dominated by natural variability, if we consider records for space-time averaged rainfall then the influence of climate change is more apparent. For example, for UK average winter mean rainfall (Supplementary Figure S19), climate change leads to a 70% increase in the occurrence of records, for 100 years of data and 12 ensemble members.

## Discussion

Our findings contrast with a common perception of climate change as more rainfall extremes as a steadily increasing trend, and more record exceedances through time even at regional scale. Instead, we expect periods of rapid change—with records being broken, some by a considerable margin – and periods when there is a pause, with no new records set. This is simply a reflection of the complex interplay between natural variability and the underlying trend in the occurrence and intensity of extreme precipitation through time. This has important implications for the communication of climate change and the prospects for society to adapt.

This analysis shows that considerable caution is needed before linking extreme rainfall events to climate change. Periods of rapid local rainfall intensification should not be taken as evidence for accelerated climate change, since the future rate of change cannot be directly inferred from them. Likewise, periods of little change when no new records are set, are not evidence that climate change is not happening. These messages need to be clearly communicated to the public. Individual record-breaking precipitation events are a manifestation of chaotic internal variability and, based on the analysis here, they continue to randomly occur over the coming decades. Nevertheless, when accumulated over many regions, the influence of climate change is more apparent. With the benefit of multiple realisations, we can see that on average climate change leads to 20% more UK-wide and 40% more regional records of local hourly rainfall. For the high emission scenario (RCP8.5), the probability of extreme hourly downpours is projected to increase by about a factor of 4, which is considerably larger than suggested by previous generations of coarser-resolution climate models. For every degree of regional warming, we find the intensity of extreme hourly downpours increases by 5–15% and there are on average almost nine more extreme events per year across the UK. Thus, changes in extreme downpours are expected to be pronounced even in intermediate to low emissions scenarios.

The results here suggest that there will be years with lots of extreme rainfall events, that may compound in terms of impacts on society. Flood impacts from heavy rainfall can be exacerbated by already wet soils, multiple regions being affected simultaneously or increased vulnerability due to recent exposure. Such extreme years

may also cluster but are unlikely to persist before there is a less extreme year. For example, in the standard member (illustrative of the variability in a single realisation), there seems to be a shift to a new state with more events exceeding 20 mm/h from 2040 (Fig. 3d). This shift is realised as a large jump in the number of events, with 80+ events in two consecutive years, followed by a period of ~10 years with fewer events, before a subsequent peak. In other members, similar features are seen although at different times. The time between extreme years varies and reflects the fact that natural variability occurs on a range of timescales, both annual and decadal. For example, the North Atlantic Oscillation and the Atlantic Multi-decadal Oscillation both have a strong influence on European climate and have been shown to dominate current observed trends in hourly rainfall over the UK[12]. An analogy for this is waves coming up a beach on an incoming tide. The tide is the long-term rising trend, but there are periods when there are larger waves, followed by lulls. The pauses in the intensification of rainfall extremes can perhaps be viewed as 'borrowed time', but they nevertheless may give opportunities for ecosystems and communities to adapt. On the other hand, this behaviour is very challenging for (re)insurances, risk management or planning, as climate information based on several decades of past observations may not be representative for the following decades.

Results here are for the UK, however, we expect similar behaviour in many other extratropical regions. Understanding the range of plausible outcomes for individual years through time is important to identify adaptation needs. Here, we provide this information for rainfall at local and hourly scales, important to allow urban planners, local authorities and flood management practitioners to plan for the future. This requires km-scale ensemble projections and suggests that similar experiments would be meaningful in other regions to understand how such potentially impact-relevant events change.

## Methods
### Models and observational data
The convection-permitting model (CPM) ensemble used here consists of twelve members at 2.2 km resolution over the UK. The simulations span December 1980 to November 2080 and use the high emissions RCP8.5 scenario. Data for three 20-year periods (1981–2000, 2021–40, and 2061–80) were released in July 2021 as part of the UK Climate Projections (UKCP) project, the so-called UKCP Local projections[20,47]. Here, for the first time, we use an augmented dataset, also including data for the intervening 20-year periods 2001–2020 and 2041–2060, resulting in a continuous 100-year timeseries of rainfall, for each ensemble member. The continuous timeseries was created by stitching together the five 20-year periods, with a 1-year period at the start of each 21-year simulation discarded as spin-up.

The 2.2 km CPM is based on a recent operational weather forecast configuration of the Met Office Unified Model (Regional Atmosphere and Land Version 1 midlatitude configuration[48], HadREM3-RA11M). Details of the 2.2 km HadREM3-RA11M model physics and experimental set-up are provided in ref. [20], with a summary of the key points provided here. At 2.2 km grid spacing, the convection scheme is switched off entirely, with convection represented explicitly on the model grid. It is termed convection-permitting because larger storms are represented, but smaller showers and convective plumes are not resolved at 2.2 km. As a result, individual updraughts are too wide and have insufficient turbulent mixing, and this leads to heavy rainfall tending to be too intense in CPMs[19]. Nevertheless, CPMs give a much more realistic representation of rainfall than traditional coarser-resolution climate models, allowing us to provide credible projections of future changes in hourly rainfall extremes.

The 2.2 km ensemble spans the UK and is driven by an ensemble of twelve regional climate model (RCM) simulations at 12 km resolution, which span Europe, and are also analysed here. The 12 km RCM ensemble (sampling different versions of the same RCM, namely the

Met Office Unified Model configuration HadREM3-GA705, which is close to the GA7 configuration[49]) is in turn driven by an ensemble of the 60km Hadley Centre global coupled model (HadGEM3-GC3.05[50]). The CPM and RCM are limited area atmosphere-only models, with daily sea surface temperature and sea ice cover prescribed from the driving global simulation. The CPM and RCM share the majority of their main physical components, but there are some notable differences. In particular, the convection scheme is switched off in the CPM, the models use different cloud schemes and only the CPM includes prognostic graupel, which is a second category of ice with higher fall speeds found in convective cloud (for more details see ref. [20]).

Different members of the RCM (and driving global model) ensemble are created by perturbing uncertain parameters in the model physics. However, this is not the case in the CPM ensemble, where the same version of the HadREM3-RA11M model is used for each ensemble member. Perturbations were not applied to the CPM, as differences in model parameterisations between the convection-permitting and coarser-resolution climate configurations of the model make it difficult to match perturbations in the driving RCM. The CPM ensemble, therefore, samples uncertainty in the driving model physics, alongside uncertainty due to natural climate variability, however, it does not sample uncertainty in the representation of local processes within the CPM itself. We note that CPM ensemble spread is actually found to be dominated by natural variability for future changes in hourly precipitation extremes, whilst model parameter uncertainty dominates for the RCM[51]. This may reflect in part the fact that parameter perturbations have not been applied to the CPM. Additionally, both the CPM and RCM ensembles only downscale versions of the Met Office Hadley Centre climate model, which has a relatively high climate sensitivity, and do not downscale other international climate models. Therefore, we may expect the CPM ensemble (and also the RCM ensemble) to provide an underestimate of uncertainty in future changes. Nevertheless, the CPM ensemble analysed here represents a major step forward, since it is the first ensemble of fully transient CPM simulations allowing us to examine the evolution of local hourly precipitation extremes over the coming decades.

For validation of the model hourly precipitation, we use the 1km-gridded hourly precipitation observations from CEHGEAR1hr[29] from Jan 1990 to Dec 2014. This dataset spans Great Britain and is based on hourly gauge observations, but also disaggregated daily gauges where there are no hourly gauges in reasonable proximity. In total there are 1900 hourly rain gauges, but the actual number on any given day ranges between 295 and 1372 due to missing data or quality issues. Disaggregation of the daily UK Centre for Ecology and Hydrology Gridded Estimates of Areal Rainfall (CEHGEAR) daily gridded dataset is performed where the nearest hourly gauge is more than 50km away, and uses a design storm profile to estimate hourly values from the daily total. It should be noted that gauge observations are affected by systematic measurement under-catch (typically about 20%, due to snow, wind blow losses and gauge exposure[30]), they may miss localised events and tend to be sited in valleys rather than at the tops of mountains. Therefore, we may expect the gauge observations to underestimate the intensity of precipitation extremes, with observational biases particularly large over mountains. For the hourly dataset, the disaggregation step leads to further potential observational bias (which could be an over or underestimation of true values depending on the actual hourly rainfall profile), especially in Scotland and south-west England where there is low hourly gauge density.

Extensive validation of the UKCP Local simulations (with observations for the historical period) was carried out to establish their credibility for projecting future change[20]. This included evaluation of UKCP Local hourly precipitation by comparison with CEHGEAR1hr. Although we do not expect correspondence between the modelled and observed values on a day-by-day, or for annual maximum values on a year-by-year, basis, with the different ensemble members giving

different realisations of natural variability, we would expect the accumulated statistics of hourly precipitation to be captured. Analysis showed that the 2.2 km CPM is able to provide a much better representation of the hourly and daily precipitation characteristics compared to the 12 km RCM, providing confidence in its ability to project future changes in local weather extremes[20]. We note that the maximum hourly value in the CEHGEAR observations for rainfall regridded to the 12 km scale is 63 mm/h. Even allowing for gauge under-catch of about 20% this would not give values above 100 mm/h, which are seen in the RCM and judged unphysical.

### Analysis approaches

In this analysis, hourly precipitation data from the models and observational dataset is regridded to a common 12 km grid before calculation of metrics to allow a fair comparison. We consider a given year as running from December to the following November, to avoid breaking the winter season in two. As a result, we have exactly 100 years for each model ensemble member (1981–2080). For the observations we have 24 whole years and 24 winters (1991–2014), but 25 summers (1990–2014).

Changes in the intensity of hourly precipitation extremes are examined through analysis of the regional annual or seasonal maximum of hourly 12 km precipitation. This corresponds to the highest value of hourly precipitation at any 12 km grid box in a given season or year, for a given region.

In the event analysis, we allow a maximum of one event per day per UK subregion to account for spatiotemporal correlation in the rainfall data i.e., multiple exceedances of the threshold on the same day within a given region are counted as one event. UK subregions are shown in Fig. 4. We considered a range of thresholds from 5mm/h to 30 mm/h to define an event (Supplementary Figure S4). An accumulation threshold of 30 mm/h is used by the UK Met Office/Environment Agency Flood Forecasting Centre as guidance to indicate likely flash flooding. However, this corresponds to rainfall over a very localised area or at a point station. For rainfall aggregated over a 12 km-by-12 km grid box, analysed here to allow comparison with the RCM, exceedance of 30 mm/h is rare (only an average of 1.4 events per year across the whole of the UK in the CPM in the 1980s, increasing to 8.8 events per year in the 2070s). Therefore, we have focussed on 20 mm/h (with results for 10 mm/h shown in Supplementary Table S1) to allow a more robust assessment of changes while still focussing on events that can potentially produce serious damage through flash floods.

We use the running 10-year mean, averaged across all 12 ensemble members, as a measure of the underlying climate change signal (forced response). This yields a relatively smooth time-varying field, where the influence of natural variability on both short and multi-decadal timescales is largely removed. This is the case because different ensemble members are not in phase in their different realisations of variability, including on multi-decadal timescales. Detrended data, which corresponds to the natural year-to-year variability in rainfall extremes (or temperature), is calculated by subtracting the underlying climate change signal from the yearly value for each ensemble member. We note that taking the multi-member mean has the advantage of reducing any influence from variability on multi-decadal timescales on the extracted climate change signal, however, it does also average across the effects of the different parameter perturbations in the driving RCM. Since RCM parameter perturbations have a relatively weak influence on future changes in the CPM[51], this is valid, with the underlying climate change signal expected to be the same across CPM members.

## Data availability

The UKCP Local data for 1981-2000, 2021-40 and 2061-80 are publicly available from the NERC EDS Centre for Environmental Data Analysis (CEDA) archive with the identifier UKCP18 Convection-Permitting Model Projections for the UK at 2.2 km resolution [http://catalogue. ceda.ac.uk/uuid/ad2ac0ddd3f34210b0d6e19bfc335539]. UKCP Local data for the intervening 20-year periods are not yet publicly available, but will be uploaded to CEDA at the time of public launch. Until then the data are available from the corresponding author upon request and are used subject to Met Office licence conditions. The UKCP Regional data are publicly available from the CEDA archive with the identifier UKCP18 Regional Climate Model Projections for the NW Europe Region [http://catalogue.ceda.ac.uk/uuid/ 45b332cd72c14fb3beddb4bf05077c97].

## Code availability

Analysis code used in this paper is available from the corresponding author upon request.

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

## Acknowledgements

We gratefully acknowledge funding from the Joint U.K. BEIS/Defra Met Office Hadley Centre Climate Programme (GA01101, E.J.K. and C.J.S.), from the EU Horizon 2020 Project XAIDA (grant agreement 101003469, E.M.F.) and by the Swiss National Science Foundation (grant 200020_178778, E.M.F.). Thanks to Simon Tucker for running the UKCP Regional projections analysed in this study.

## Author contributions

E.J.K. wrote the paper, carried out the analysis and led the delivery of the UKCP Local Projections. E.M.F. advised on the analysis and co-wrote the paper. C.J.S. carried out the UKCP Local simulations and commented on the paper.

## Competing interests

The authors declare no competing interests.
