## [Peer Review File · Nature Communications]

Variability conceals emerging trend in 100yr projections of UK local hourly rainfall extremesREVIEWER COMMENTS

Reviewer #1 (Remarks to the Author):

This manuscript used the ensemble of convection-permitting transient simulations to examine the emerging signal in precipitation extremes at local and hourly scales over 100 years. The influence of climate change on the occurrence of hourly precipitation records was also examined. They found that the intensity of extreme hourly downpours increases at 5-15%/K and the occurrence of regional hourly rainfall records is 40% larger than in the absence of warming. The hourly precipitation extremes (exceeding 20mm/h) are 4 times as frequently by the 2070s. These results can provide more important information for the policy-maker in the future. I recommend accepting it after a major revision:

My major revisions are:

- 1. More performance of CPM in simulating the hourly precipitation extremes should be evaluated, including the pattern of hourly precipitation extremes. This may give more information about the ability of CPM to represent the characteristics of hourly precipitation extremes.**
- 2. Please give the explanation of "any grid point" in your caption of Figure 2 (L152). How you select any grid point.**
- 3. As for the increasing of interannual variability with time in L114-115, the reference about the potential mechanism should be cited to support it.**
- 4. In Line 184-185. The authors said that due to representing the local dynamical feedbacks of CPM, higher CC scaling values were found in some regions/seasons. The scaling values are higher for winter than in summer, especially in the southern region. This means that the local dynamical feedback is stronger in winter than in summer. Could you give the reference or the explanations to support this? Whether the phenomenon with higher scaling values in winter than in summer can be represented in RCM?**
- 5. Why did the authors select the 20mm/h as the threshold of precipitation extremes? The percentile-based threshold should be added to demonstrate the robustness of the results.**
- 6. It is interesting that for the annual extremes, scaling values are consistent between the raw and smoothed data for the intensity and frequency metrics in CPM (Figure 4). A quite large difference was found for RCM (Table S2), this should highlight in your article to demonstrate the advantage of CPM.**

Reviewer #2 (Remarks to the Author):

Review of "Variability conceals emerging trend in 100yr projections of local hourly rainfall extremes" by Kendon et al.

The authors present a study of changes in hourly rainfall extremes, using an initial conditions ensemble of transient simulations with a convection permitting RCM (CPM) over the UK. They compare the present day representation of extreme precipitation as well as simulated changes with simulations from a standard-resolution RCM. The authors find a much more realistic representation by the CPM, a strong increase in intensities, and a major role of internal variability in producing record high extreme events.

While the results are all sound and relevant, I have a couple of major concerns that in my view question publication in a high impact letter format such as Nature Communications.

Major Concerns

First of all, the paper lacks focus and sometimes depth. While the abstract only refers to the findings of the CPM, a substantial part of the actual manuscript includes comparisons with a standard-resolution RCM. Why including this assessment of potential added value if is not interesting enough to include it in the abstract? Also, because of the limited space, the comparison is sometimes rather shallow. E.g., the authors in two instances refer to the possibility that RCM biases/implausible changes result from grid-box storms (line 89 and 267), but only speculate rather than assessing this point in detail. Removing this intercomparison would help reducing the number of supplementary figures.

Second, I am wondering about the novelty of the paper. Indeed, I am not aware of any other transient climate change simulations with CPMs, which would allow for studies of record behaviour and signal emergence. But beyond this point, the study is simply state-of-the-art. In fact, the authors are silent about a series of important papers on similar topics. Coppola et al. (Clim. Dynam., 2021) presented the first ensemble of (10-year time-slice) convection permitting simulations over Central Europe, Ban et al. (Clim. Dynam, 2021) evaluated this ensemble for hourly rainfall extremes (including a comparison with standard-RCMs) and Pichelli et al. (Clim. Dynam., 2021) assessed future changes of hourly precipitation extremes in this ensemble. None of these papers has been cited, even though the first author of this manuscript is co-author on all of them.

Finally, also regarding other topics the cited literature is often rather selective. The title and abstract both use the word "emerging", but this term is not used in the main text anymore. Given the discussion of waiting times between record breaking events being influenced strongly by internal variability, the term "emergence" makes sense of course. But then I would suggest to frame the whole paper more along these lines. This would require adding a short paragraph in the introduction about the concept (in particular referring to the term "time of emergence") and then adding relevant literature, i.e., papers on emergence of rainfall trends, in particular extremes: Maraun, Env. Res. Lett., 2012; King et al., Env. Res. Lett., 2015; Aalbers et al., Clim. Dynam. 2018. See also IPCC AR6 WG1 Chapter 10 Section 10.4.3.2 for a discussion and further references. Also the whole paper should be put into the context of observed changes in extreme rainfall, both globally and within the UK, and both at the daily and sub-daily scale. Some example references are Alexander et al., J. Geophys. Res., 2006 (and/or updates), Maraun et al., Int. J. Climatol., 2008; Ye et al., Science Adv., 2017; Cotteril et al., Wea. Clim. Extr. 2021.

Further comments

p1, l 27: these floods are not a good motivation for changes in hourly precipitation. The affected river basins have been of a size that responds more to daily precipitation. Instead, the examples of Urban flooding mentioned later in the paper would be more sensible.

Does the aggregation to the whole of the UK (e.g. Fig S10) make sense? The records will be dominated by the regions with highest precipitation. Would it make sense to normalise the data? I suggest to add a short discussion.

Overall assessment

In its current version, I believe the manuscript is too unfocused for a letter format, and not novel enough to merit publication in a high impact journal such as Nature

Communications. I would rather suggest to expand the intercomparison with the standard RCM, to put the paper into the broader context of important scientific developments in the fields of signal emergence, trends of extreme (also hourly) precipitation and convection permitting simulations, and then to submit a considerably longer manuscript to a disciplinary journal such as Climate Dynamics or maybe Communications Earth & Environment.

Reviewer #3 (Remarks to the Author):

The authors present a comprehensive assessment of extreme rainfall events over the United Kingdom using an ensemble of high-resolution simulations in historic and future climates. They show trends in the magnitude and frequency of heavy rainfall events and quantify the uncertainty due to internal variability. They also show how the magnitude of heavy rainfall events scale with local temperature increases and find that the Clausius-Clapeyron relationship is not always followed and can vary in different locations and different time periods. They also show that there are temporal variations in the number of records broken over time, with large gaps in record-breaking events, however, with a higher number of records still being broken due to climate change enhancement.

Overall, this paper makes a useful contribution to the climate literature. It makes use of a rare type of climate dataset: a high-resolution, convection-permitting, almost century-long medium-sized ensemble. This type of dataset, though computationally intensive, can uncover the role of climate change and climate variability on change in rainfall on very local scales, and can better capture the processes that lead to these types of events. A coarser model or a smaller ensemble would not be able to do that. The analysis that the authors carry out is sound and interesting.

However, there are points where the authors did not take the opportunity to really let this dataset and analysis shine (see comments about Figures 3 and 5). Additionally, there are areas in the text where the authors make conclusions that are not well-backed, and where methods are unclear. Therefore, I recommend that the paper be returned for revisions. Below, I outline my comments and suggestions.

Lines 68-70: Please give brief descriptions of the CPM and RCM model and simulation set ups here. I know they are in the methods, but I think this is important context for the results you are presenting here.

Line 71: Please give a brief description of this dataset.

Line 73: In the methods, the authors state that some large rainfall events may not be represented in the observational dataset. I am curious to know if these omissions would be close to some of the values that the RCM seems to be overestimating?

Line 88-90: I think this sentence warrants some elaboration especially for a journal with a broader audience. Could you please explain "unphysical grid point storms" a little further?

Line 90: How do you know that these values are "unphysical" and not just due to the lack of representation of internal variability in the observations? Please elaborate.

Figure 1 panel d. Please indicate which box plots correspond to which label on the y-axis. From the current labels, it is not clear.

Line 113: Please note here that this is a visual inspection and not a calculated trend.

Line 138: Could you please define "lots" of extreme rainfall events in a more specific and quantitative way? It seems a bit hand-wavy right now.

Line 141: Typo - m/h  mm/h

Figure 3: Why do the authors not show grid-point values of these values and ratios on three maps, with each grid points shading representing the ensemble-average (1) historic and (2) future values, and the (3) ratio? Given the power of the CPM simulations to allow for hyper-local information, I think the authors are missing a big opportunity to visually show the importance of their fine scale runs and the localization of extreme rainfall events.

Figure 3: Some of the text on the map is difficult to read. Suggestion to change colors to improve readability, or to separate the bar plots from the map.

Line 170: Please specify that this is degree Kelvin.

Lines 199-202: Could you please elaborate on this conclusion? I'm not sure I am able to easily connect the dots between the scatter and the importance of sub-seasonal variability.

Figure 5: Like my comment about Figure 3, (1) readability of some the text needs to be improved, and (2) can the authors show the grid-point level data for the scaling coefficients?

Line 267: Could you please elaborate on why this is not realistic? Also, please clarify what you mean by "grid point storms".

Line 311: Typo: no evidence  not evidence

Line 317: Please specify which high emission scenario (i.e., RCP8.5).

Line 365: Please clarify that you are using twelve simulations from the SAME RCM and not simulations from twelve DIFFERENT RCMs.

Lines 378-381: I am not sure that your results showing the ensemble spread ONLY reflect internal variability as stated in the main text but may also reflect model physics uncertainty. This is because I am assuming the boundary conditions for each ensemble will be different because of the different physics parameterizations of the driving model. In that case, I recommend additional explanation or reasoning in terms of what is reflected in the spread of the ensemble.

Reviewer response document

Please find below detailed responses to the reviewers' comments, with responses in red text. Also uploaded is a manuscript file with revisions marked as tracked changes.

Reviewer #1 comments

This manuscript used the ensemble of convection-permitting transient simulations to examine the emerging signal in precipitation extremes at local and hourly scales over 100 years. The influence of climate change on the occurrence of hourly precipitation records was also examined. They found that the intensity of extreme hourly downpours increases at 5-15%/K and the occurrence of regional hourly rainfall records is 40% larger than in the absence of warming. The hourly precipitation extremes (exceeding 20mm/h) are 4 times as frequently by the 2070s. These results can provide more important information for the policy-maker in the future. I recommend accepting it after a major revision:

Thank you for your time taken to review the paper. Your comments have been very helpful in improving the paper.

My major revisions are:

1. More performance of CPM in simulating the hourly precipitation extremes should be evaluated, including the pattern of hourly precipitation extremes. This may give more information about the ability of CPM to represent the characteristics of hourly precipitation extremes.

Maps showing the mean and standard deviation of annual maximum values at every 12km grid box across the UK, for the 24-year observational period (1991-2014) have now been added to the paper (Fig 2, copied below). In the paper, the figure compares the observed realization with the CPM and RCM ensemble median value, however, below (Fig R1) we also show the result for the standard CPM and RCM member indicative of an individual model realization. The CPM and RCM ensemble medians give a smoother field than the CEHGEAR observations, as expected given the latter is a single realization. In general, the CPM gives better agreement with the observations than the RCM in terms of the mean and variability in local annual maximum hourly precipitation at the 12km grid box scale. The tendency for the RCM to overestimate regional maxima is less apparent for local maxima, where the annual maximum values locally can be considerably less than those regionally. However, the variability of local annual maximum values is consistently overestimated in the RCM, except over northern Scotland. Over northern Scotland, both models underestimate annual maximum hourly precipitation, with the CEHGEAR observations suggesting much higher values over high ground in the north and west. However, it should be noted that we have less confidence in the observational

values in this region due to a lack of availability of hourly gauges. In particular, the hourly gauge density is low over Scotland, with the distances to the nearest gauge greatest over western Scotland (Lewis et al 2018) [ref 29] and hence for these locations, where there are no hourly gauges in the local vicinity, hourly precipitation values are estimated by disaggregating daily gauge data using an average storm profile. In regions of orographic rainfall, this process may feasibly lead to peak values being overestimated. Text discussing this has been added to the Results section.

Fig 2. Present-day mean and variability in local annual maximum hourly precipitation. 24-year (top) mean and (bottom) standard deviation of annual maximum precipitation at every 12km grid box, for 1991-2014. Results are shown for (left) CEHGEAR observations and the median of the (centre) CPM and (right) RCM ensembles.

Fig R1. As Fig 2 now shown in the main paper, but showing the standard member realization of the CPM and RCM, instead of the ensemble median.

To give a view of how the CPM simulates less extreme values of hourly precipitation and its seasonal variation, maps showing the mean and 99.95th percentile of hourly precipitation at the 12km scale for the present-day have been added to the supplementary material (Supp Figs S5-6, copied below). Biases are consistently lower for the CPM compared to the RCM, and the CPM shows root mean square (RMS) errors consistently less than observational uncertainty (as measured by the difference between the CEHGEAR gauge data and NIMROD radar observations). This provides evidence that the CPM is able to capture both seasonal mean and heavy hourly precipitation, although there is some suggestion that summertime heavy rainfall is too heavy in the CPM compared to both observational datasets.

Supp Fig S5. **Model performance in winter.** (top) Seasonal mean precipitation and (bottom) 99.95th percentile of hourly precipitation (mm/h), in DJF, for (left) CEHGEAR (1990-2014) and differences (%) with respect to the (centre left) radar (2003-2017), (centre right) RCM and (right) CPM ensemble mean for the baseline climate (1981-2000). The UK average values (in mm/h) and root mean square errors (RMS, in %) are indicated.

Supp Fig S6. **Model performance in summer.** (top) Seasonal mean precipitation and (bottom) 99.95th percentile of hourly precipitation (mm/h), in JJA, for (left) CEHGEAR (1990-2014) and differences (%) with respect to the (centre left) radar (2003-2017), (centre right) RCM and (right) CPM ensemble mean for the baseline climate (1981-2000). The UK average values (in mm/h) and root mean square errors (RMS, in %) are indicated.

Annual timeseries plots showing performance of the CPM and RCM compared to observations in representing regional annual maximum values, for UK sub-regions in addition to those shown in Fig 1, have been added to the supplementary material (Supp Fig S3, and copied below). This shows that the CPM underestimates regional maximum values of hourly precipitation over regions in Scotland, but note comment above regarding lower reliability of observational values here. Over regions in England and Wales, the CPM shows good agreement with the observations in terms of the regional maximum values and their year-to-year variability – and consistently better performance than the RCM, which overestimates values and shows some physically implausible hourly accumulations of >100mm/h at the 12km scale. This discussion has been added to the text.

Supp Fig S3: **Regional annual maximum hourly precipitation, for regions across the UK.** Maximum hourly precipitation (mm/h) occurring within the region (considering all 12km grid boxes) in a given year, for 1991-2014. Results are shown for CEHGEAR observations (black), and the CPM (red) and RCM (blue), for the standard member (thin line), ensemble mean (thick line) and ensemble min-max range (shaded). Regions are as shown in Fig 4.

Reference

Lewis, E. et al (2018) A rule based quality control method for hourly rainfall data and a 1 km resolution gridded hourly rainfall dataset for Great Britain: CEH-GEAR1hr, *Journal of Hydrology*, 564, pp 930-943, <https://doi.org/10.1016/j.jhydrol.2018.07.034>.

2. Please give the explanation of “any grid point” in your caption of Figure 2 (L152). How you select any grid point.

This term was used in the captions of both Fig 1 and original Fig 2 (now Fig 3). It refers to selecting the maximum hourly precipitation occurring within the region (considering all 12km grid boxes) for a given year (considering all hours in that year). This has been clarified in the figure captions.

3. As for the increasing of interannual variability with time in L114-115, the reference about the

potential mechanism should be cited to support it.

The interannual variability in regional annual maximum values is shown in Fig 3(c) (and also Supp Fig S7c and S7d), along with the running 10-year multi-member mean. We have additionally plotted the ratio of the standard deviation to the mean in the figures below (Figs R2 and R3), which confirms that the standard deviation largely increases in proportion to the mean (i.e. there is no evidence of an upward trend in the ratio). Thus, the increase in interannual variability in annual maxima is largely simply a reflection of the increase in extreme hourly precipitation through time. However, there is some evidence that interannual variability in extremes is relatively higher for some periods – e.g. 2040s and 2050s over SE England in the CPM, but this is not sustained. It is likely that such features relate to phases of multi-decadal variability that make it difficult to discern any underlying trend in an individual realisation (Kendon et al 2018) [ref 12]. We note that the UKCP18 global simulations under RCP8.5 display a strong Atlantic Meridional Overturning Circulation (AMOC) weakening, which translates into increased storminess (Jackson et al 2015), and also in the 2040s the Arctic becomes seasonally ice free, with possible (but uncertain) links between Arctic amplification and midlatitude severe weather (Cohen et al 2020). However further work, beyond the scope of this paper, is needed to identify if any such mechanisms lead to increased interannual variability common to all ensemble members.

This is now discussed in the text, and the following references have been added:

Jackson, L.C., Kahana, R., Graham, T. et al. Global and European climate impacts of a slowdown of the AMOC in a high resolution GCM. *Clim Dyn* 45, 3299–3316 (2015).

<https://doi.org/10.1007/s00382-015-2540-2>

Cohen, J., Zhang, X., Francis, J. et al. (2020) Divergent consensus on Arctic amplification influence on midlatitude severe winter weather. *Nat. Clim. Chang.* 10, 20–29. <https://doi.org/10.1038/s41558-019-0662-y>

Figure R2: Variability in regional annual maximum hourly precipitation, for South East England. (left) Variability in regional annual maximum values in running 10-year window, and for the models additionally across the 12 ensemble members [As Supp Fig S7c]. Solid line shows the standard deviation (mm/h) of yearly values, and dashed line the smoothed (multi-member) 10-year running mean (mm/h, divided by 2). (right) Ratio of standard deviation to running 10-year mean. Results are shown for the CPM (red), RCM (blue) and CEHGEAR observations (black).

Figure R3: Variability in regional annual maximum hourly precipitation, for North West England. (left) Variability in regional annual maximum values in running 10-year window, and for the models additionally across the 12 ensemble members [As Supp Fig S7d]. Solid line shows the standard deviation (mm/h) of yearly values, and dashed line the smoothed (multi-member) 10-year running mean (mm/h, divided by 2). (right) Ratio of standard deviation to running 10-year mean. Results are shown for the CPM (red), RCM (blue) and CEHGEAR observations (black).

4. In Line 184-185. The authors said that due to representing the local dynamical feedbacks of CPM, higher CC scaling values were found in some regions/seasons. The scaling values are higher for winter than in summer, especially in the southern region. This means that the local dynamical feedback is stronger in winter than in summer. Could you give the reference or the explanations to support this? Whether the phenomenon with higher scaling values in winter than in summer can be represented in RCM?

Higher scaling rates in winter than in summer are consistent with the greater increase in moisture availability in winter. In summer although specific humidity increases, relative humidity decreases (Kendon et al 2010) and thus moisture availability does not increase with temperature as quickly as the temperature-dependent maximum (i.e. following the change in saturated specific humidity). Note that for the seasonal scaling analysis we scale with seasonal mean temperature, rather than annual mean temperature, so that any seasonal differences in warming are accounted for (this is now clarified in the Fig 6 caption). Greater decreases in relative humidity in the south than the north of the UK, also explain the north-south gradient in scaling rates. Thus the changes in moisture

availability with warming largely explain the seasonal and regional differences in scaling that we see in Fig 6.

In addition to changes in moisture availability, changes in weather patterns and thermodynamic stability may lead to scaling rates that depart substantially from CC (e.g. Schneider and O’Gorman, 2008; O’Gorman and Schneider, 2009; Pfahl et al. 2017; Chan et al 2016). Dynamical feedbacks caused by latent heating (and evaporation of rain causing downdrafts and cold pools) have also been cited as a possible explanation for super-CC scaling rates for hourly precipitation extremes (Lenderink et al 2021). Such local dynamical feedbacks are only captured by CPMs, and thus we may anticipate different scaling rates between the CPM and RCM. From Supp Table S2, it can be seen that there is not a consistent relationship between the scaling coefficients in the CPM and RCM. For the smoothed underlying climate change signal, scaling rates are typically higher in the RCM than CPM for annual extremes, but much more variable for seasonal extremes. These scaling rates for the RCM are not reliable, with grid point storms in the RCM likely masking any differences in scaling due to resolved local dynamical processes in the CPM.

This discussion has been added to the text along with the following references:

Kendon, E.J., Rowell, D.P. & Jones, R.G. Mechanisms and reliability of future projected changes in daily precipitation. *Clim Dyn* 35, 489–509 (2010). <https://doi.org/10.1007/s00382-009-0639-z>

Schneider, T. & O’Gorman, P. A. Moist convection and the thermal stratification of the extratropical troposphere. *J. Atmos. Sci.* 65, 3571–3583 (2008)

O’Gorman, P. A. & Schneider, T. The physical basis for increases in precipitation extremes in simulations of 21st-century climate change. *Proc. Natl Acad. Sci. USA* 106, 14773–14777 (2009).

Pfahl, S., O’Gorman, P. & Fischer, E. Understanding the regional pattern of projected future changes in extreme precipitation. *Nature Clim Change* 7, 423–427 (2017).
<https://doi.org/10.1038/nclimate3287>

Chan, S., Kendon, E., Roberts, N. et al. Downturn in scaling of UK extreme rainfall with temperature for future hottest days. *Nature Geosci* 9, 24–28 (2016). <https://doi.org/10.1038/ngeo2596>

Lenderink G. et al (2021) Scaling and responses of extreme hourly precipitation in three climate experiments with a convection-permitting model *Phil. Trans. R. Soc. A*.3792019054420190544
<http://doi.org/10.1098/rsta.2019.0544>

5. Why did the authors select the 20mm/h as the threshold of precipitation extremes? The percentile-based threshold should be added to demonstrate the robustness of the results.

20mm/h was chosen as an impacts-relevant threshold. As discussed in the Methods, 30mm/h is used by the Met Office – Environment Agency Flood Forecasting Centre as the threshold to trigger a flash flood warning, but this is for rainfall at a very local (order 1km or point) scale. We did explore such a high threshold, but for rainfall aggregated to the 12km scale such a threshold is too extreme to allow a robust analysis. Therefore, we reduced the threshold to 20mm/h for rainfall at the 12km scale, to be indicative of events that can potentially produce serious damage through flash flooding.

Results with a 10mm/h threshold are already shown in Supp Table S1, however, we have also added a figure to the supplementary material (Supp Fig S4) showing the timeseries of events with thresholds of 5mm/h, 10mm/h, 20mm/h and 30mm/h. Below we replicate this figure, and also

provide a plot zooming into the 24-year observational period (Fig R4) to allow a clearer assessment of model performance for the different thresholds.

Figure R4: Model performance in representing frequency of precipitation events exceeding a range of high thresholds. Number of events per year across the UK exceeding (a) 5mm/h, (b) 10mm/h (c) 20mm/h and (d) 30mm/h, for precipitation at 12km scale. Results are shown for CEHGEAR observations (black), and the CPM (red) and RCM (blue) for the standard member (thin line), ensemble mean (thick line) and ensemble min-max range (shaded) for 1991-2014. Threshold exceedances occurring within a UK subregion on the same day are considered part of a single event.

Supp Fig S4: Frequency of precipitation events exceeding a range of high thresholds. Number of events per year across the UK exceeding (a) 5mm/h, (b) 10mm/h (c) 20mm/h and (d) 30mm/h, for precipitation at 12km scale. Results are shown for CEHGEAR observations (black, 1991-2014), and the CPM (red) and RCM (blue) for the standard member (thin line), ensemble mean (thick line) and ensemble min-max range (shaded) for 1981-2080. Threshold exceedances occurring within a UK subregion on the same day are considered part of a single event.

The CPM shows good agreement with the observations in terms of the exceedance frequencies for all four thresholds examined. The observations typically lie within the ensemble spread. This is not true for the RCM which underestimates the number of events exceeding 5mm/h and overestimates the number of events exceeding 20mm/h and 30mm/h.

In terms of the future change results, Supp Table S1 compares future changes for the 10mm/h and 20mm/h thresholds. We do not show results with a 30mm/h threshold as there are only an average of 1.4 events per year across the whole of the UK in the CPM in the 1980s (increasing to 8.8 events per year in the 2070s, as stated in the Methods), which are too few to allow a robust assessment on a regional basis. From Supp Table S1 it can be seen that results for a 10mm/h threshold are similar to those for a 20mm/h threshold in the sense that future changes are consistently larger in the CPM than in the RCM and in the north of the UK than in the south. However, differences in future changes between the CPM and RCM are more marked for the higher threshold. In general, the higher the threshold the greater the percentage future change in the number of events. A note has been added to the text on the robustness of the results to the choice of threshold.

Results looking at the number of events exceeding the present-day 99.99th percentile of hourly precipitation have been added to Supp Table S1. The 99.99th percentile threshold corresponds to 9.81mm/h in the observations (1991-2010 baseline), and 9.28 mm/h in the CPM and 9.35mm/h in the RCM (1981-2000 baseline). This has been calculated by pooling data from all 12km grid cells across the UK and all hours, from the 20-year baseline climate, and in the case of the models the ensemble-mean is then calculated. From Supp Table S1 it can be seen that differences in future changes between the CPM and RCM are reduced when using the 99.99th percentile threshold. However, for almost all regions (except Northern Ireland and East of England) future increases in the number of events are still greater in the CPM than RCM. This is noted in the text.

Evaluation plots looking at the number of events exceeding percentile thresholds (99.95th and 99.99th percentiles of hourly precipitation) are shown below (Fig R5). The 99.95th percentile corresponds to 6.68 mm/h in the observations, 6.48 mm/h in the CPM and 5.74 mm/h in the RCM. It can be seen that both models agree well with the observations for both percentile thresholds. By definition we expect the same total number of grid cell hourly exceedances in the present-climate, but the number of ‘events’ may differ depending on how exceedances are clustered in space and time. For both thresholds, the CPM generally shows more independent events than the RCM, which indicates that hourly rainfall events are less clustered (i.e. occur on more separate days or subregions). Since this is a more subtle point in terms of rainfall evaluation and less user-relevant, compared to the performance in representing the exceedance of absolute thresholds, we have not included it in the paper.

Fig R5: Model performance in representing frequency of precipitation events exceeding high percentile thresholds. Number of events per year across the UK exceeding the present-day (a) 99.95th and (b) 99.99th percentile of hourly precipitation across the UK. Percentiles are calculated by pooling all 12km grid boxes and all hours, in the 20-year baseline period (1991-2010 for CEHGEAR and 1981-2000 for the models). The 99.95th percentile is 6-7mm/h depending on dataset, and the 99.99th percentile is 9-10mm/h. Results are shown for CEHGEAR observations (black), and the CPM (red) and RCM (blue) for the standard member (thin line), ensemble mean (thick line) and ensemble min-max range (shaded) for 1991-2014. Threshold exceedances occurring within a UK subregion on the same day are considered part of a single event.

6. It is interesting that for the annual extremes, scaling values are consistent between the raw and smoothed data for the intensity and frequency metrics in CPM (Figure 4). A quite large difference was found for RCM (Table S2), this should highlight in your article to demonstrate the advantage of CPM.

We have added a note to highlight this:

“The scaling coefficient itself is similar between the raw (or detrended) and smoothed data in the CPM for both the intensity and frequency metrics in Fig 5. This is also true across all regions for annual extremes in the CPM (Supp. Table S2). For the RCM, large differences in the scaling coefficients between the raw and smoothed data occur, again highlighting issues with the reliability of the RCM scaling results.”

Reviewer #2 comments

Review of "Variability conceals emerging trend in 100yr projections of local hourly rainfall extremes" by Kendon et al.

The authors present a study of changes in hourly rainfall extremes, using an initial conditions ensemble of transient simulations with a convection permitting RCM (CPM) over the UK. They compare the present day representation of extreme precipitation as well as simulated changes with simulations from a standard-resolution RCM. The authors find a much more realistic representation by the CPM, a strong increase in intensities, and a major role of internal variability in producing record high extreme events.

While the results are all sound and relevant, I have a couple of major concerns that in my view question publication in a high impact letter format such as Nature Communications.

Thank you for your time taken to review the paper. Below we address each of your concerns in turn.

Major Concerns

First of all, the paper lacks focus and sometimes depth. While the abstract only refers to the findings of the CPM, a substantial part of the actual manuscript includes comparisons with a standard-resolution RCM. Why including this assessment of potential added value if is not interesting enough to include it in the abstract? Also, because of the limited space, the comparison is sometimes rather shallow. E.g., the authors in two instances refer to the possibility that RCM biases/implausible changes result from grid-box storms (line 89 and 267), but only speculate rather than assessing this point in detail. Removing this intercomparison would help reducing the number of supplementary figures.

The paper was originally submitted for publication in Nature Climate Change (before being transferred to Nature Comms), and hence the abstract and text were deliberately short to meet the length requirements for that journal. However, now that the article is being considered for Nature Comms we have more space to add more in-depth discussion of results.

The abstract itself can only be 150 words, but we have revised this to point to one of the key differences between the CPM and RCM in terms of future changes.

In the main text we have added further results comparing the CPM and RCM, both in terms of present-day performance and future changes, for a range of thresholds. This gives more insights into the model differences, with further discussion added to the text. The fact that the CPM performs systematically better across different thresholds, and gives larger relative changes (as now highlighted in the abstract), is important. The tendency for the RCM to overestimate the occurrence of very rare extremes (20mm/h and 30mm/h thresholds) is consistent with previous work (Chan et al 2014, already cited). In particular the occurrence of unphysically high values of greater than 100mm/h at the 12km scale (identified in Fig 1) is diagnostic of a grid point storm. However, further analysis to identify grid point storms in the RCM is beyond the scope of this study. In particular actually identifying grid point storms is not straightforward as there is no clear threshold that can be applied. There will be grid point storms that lead to high totals, but which are below 100mm/h, and so even applying a 100mm/h threshold to exclude unphysical events would still leave grid point storms in the dataset. Work to use additional characteristics of grid points storms, relating very different convective fractions compared to neighbouring grid points, has also proved difficult. Thus, assessing grid point storms in the RCM would be a study in its own right, and here we would like to mainly focus on the CPM results.

Further discussion of differences in the scaling results between the CPM and RCM have been added. This includes discussion of the importance of dynamical feedbacks that are only captured in the CPM, as well as the likely role of grid point storms.

The following references have been added with respect to RCM performance and CPM added value:

Hanel, M.T. and A. Buishand (2010) On the value of hourly precipitation extremes in regional climate model simulations, *Journal of Hydrology*, Volume 393, Issues 3–4, PP 265-273, <https://doi.org/10.1016/j.jhydrol.2010.08.024>.

Hohenegger C, Brockhaus P, Schär C (2008) Towards climate simulations at convection permitting scales. *Meteorol Z* 17:383–394

Lean, H. W., Clark, P. A., Dixon, M., Roberts, N. M., Fitch, A., Forbes, R., & Halliwell, C. (2008). Characteristics of High-Resolution Versions of the Met Office Unified Model for Forecasting Convection over the United Kingdom, *Monthly Weather Review*, 136(9), 3408-3424. Retrieved Nov 3, 2022, from <https://journals.ametsoc.org/view/journals/mwre/136/9/2008mwr2332.1.xml>

Lenderink G. et al (2021) Scaling and responses of extreme hourly precipitation in three climate experiments with a convection-permitting model *Phil. Trans. R. Soc. A*.3792019054420190544 <http://doi.org/10.1098/rsta.2019.0544>

Second, I am wondering about the novelty of the paper. Indeed, I am not aware of any other transient climate change simulations with CPMs, which would allow for studies of record behaviour and signal emergence. But beyond this point, the study is simply state-of-the-art. In fact, the authors are silent about a series of important papers on similar topics. Coppola et al. (Clim. Dynam., 2021) presented the first ensemble of (10-year time-slice) convection permitting simulations over Central Europe, Ban et al. (Clim. Dynam, 2021) evaluated this ensemble for hourly rainfall extremes (including a comparison with standard-RCMs) and Pichelli et al. (Clim. Dynam., 2021) assessed future

changes of hourly precipitation extremes in this ensemble. None of these papers has been cited, even though the first author of this manuscript is co-author on all of them.

This paper is the first time that transient 100y climate projections are available for a CPM. This is a key step forward in national climate capability as it allows us to identify how changes in local hourly rainfall extremes (important for flooding and only captured by CPMs) will manifest through time. Previous studies based on traditional climate models looking at trends in rainfall extremes have largely focussed on daily rainfall (e.g. Maraun, 2013 now referred to in the paper). The CPM provides a much more reliable representation of hourly extremes, projecting a much stronger increase in the frequency of events. It is an important finding of this study that the frequency changes in extreme hourly precipitation are systematically higher than in a more traditional RCM. The new setup also allows us to better put observed events into the context of climate change. It provides robust estimates of change in precipitation extremes per degree of warming and new insights into record behaviour. Particularly, the fact that extreme precipitation does not evolve gradually but includes multiple decades of few events followed by a cluster of very extreme events is important information for many stakeholders. Therefore, this study represents a significant advance in impacts-relevant research.

We have added references to papers exploiting the first multi-model ensemble of CPMs, which we agree is also a major step-forward in climate modelling capability. These multi-model simulations are all relatively short time-slice simulations (so do not provide a view of the emerging changes in extremes through time, which is novel to our study) but allow a more comprehensive assessment of uncertainties in future changes in local rainfall extremes. Thus these different approaches complement each other, and ultimately our ambition in the regional modelling community should be to combine these. However, given their high computational cost, fully transient multi-model ensemble projections at km-scale, requiring coordination across many different modelling groups, are some way off.

Given the additional space available for a Nature Comms article, we have added text to the Introduction providing more discussion of the state-of-the-art in convection-permitting climate modelling. Thank you for providing additional references, we have now added these.

New references added:

Maraun D. (2013) When will trends in European mean and heavy daily precipitation emerge? *Environ. Res. Lett.*, Volume 8, Number 1, <https://doi.org/10.1088/1748-9326/8/1/014004>

Coppola, E. et al (2018) A first-of-its-kind multi-model convection permitting ensemble for investigating convective phenomena over Europe and the Mediterranean. *Climate Dynamics*. <https://doi.org/10.1007/s00382-018-4521-8>

Ban, N. et al (2021) The first multi-model ensemble of regional climate simulations at kilometer-scale resolution, Part I: Evaluation of precipitation. *Clim Dyn.* <https://doi.org/10.1007/s00382-021-05708-w>

Pichelli, E. et al. (2021) The first multi-model ensemble of regional climate simulations at kilometer-scale resolution Part 2: Historical and future simulations of precipitation. *Clim Dyn.* <https://doi.org/10.1007/s00382-021-05657-4>

Kendon, E. J., N. M. Roberts, C. A. Senior, M. J. Roberts (2012) Realism of rainfall in a very high resolution regional climate model, *J Climate*, 25, 5791-5806

Finally, also regarding other topics the cited literature is often rather selective. The title and abstract both use the word "emerging", but this term is not used in the main text anymore. Given the discussion of waiting times between record breaking events being influenced strongly by internal variability, the term "emergence" makes sense of course. But then I would suggest to frame the whole paper more along these lines. This would require adding a short paragraph in the introduction about the concept (in particular referring to the term "time of emergence") and then adding relevant literature, i.e., papers on emergence of rainfall trends, in particular extremes: Maraun, *Env. Res. Lett.*, 2012; King et al., *Env. Res. Lett.*, 2015; Aalbers et al., *Clim. Dynam.* 2018. See also IPCC AR6 WG1 Chapter 10 Section 10.4.3.2 for a discussion and further references. Also the whole paper should be put into the context of observed changes in extreme rainfall, both globally and within the UK, and both at the daily and sub-daily scale. Some example references are Alexander et al., *J. Geophys. Res.*, 2006 (and/or updates), Maraun et al., *Int. J. Climatol.*, 2008; Ye et al., *Science Adv.*, 2017; Cotterill et al., *Wea. Clim. Extr.* 2021.

A discussion of observed trends and the emergence of changes in local rainfall extremes compared to natural variability has been added to the Introduction. In the exceedance of records through time section, we point to the relevance of this analysis for the question of emergence and briefly discuss implications for the emergence time of changes in local rainfall extremes.

Thank you for the suggestions of additional references. The following have been added:

Maraun D. (2013) When will trends in European mean and heavy daily precipitation emerge? *Environ. Res. Lett.*, Volume 8, Number 1, <https://doi.org/10.1088/1748-9326/8/1/014004>

Maraun, D., T. J. Osborn, N. P. Gillett (2008) United Kingdom daily precipitation intensity: improved early data, error estimates and an update from 2000 to 2006, *Int. J. of Climatol.* 28 (6), pp 833-842, <https://doi.org/10.1002/joc.1672>

Brown S. J. (2018) The drivers of variability in UK extreme rainfall, *Int. J. of Climatol.* 38 (S1), pp119-130, <https://doi.org/10.1002/joc.5356>

Fowler, H.J., Lenderink, G., Prein, A.F. *et al.* Anthropogenic intensification of short-duration rainfall extremes. *Nat Rev Earth Environ* 2, 107–122 (2021). <https://doi.org/10.1038/s43017-020-00128-6>

Cotterill D., P. Stott, N. Christidis, E. Kendon (2021) Increase in the frequency of extreme daily precipitation in the United Kingdom in autumn, *Weather and Climate Extremes*, Volume 33, <https://doi.org/10.1016/j.wace.2021.100340>.

Further comments

p1, l 27: these floods are not a good motivation for changes in hourly precipitation. The affected river basins have been of a size that responds more to daily precipitation. Instead, the examples of Urban flooding mentioned later in the paper would be more sensible.

We agree that the floods in central Europe were as a consequence of high daily totals. However, as a very extreme recent flooding event, where the intensity of rainfall has been formally linked to climate change, we think it is still useful to mention this. Such an event and its attribution to climate

change really raises the public awareness of changing risks from surface water flooding. We have now emphasized this point in the paper.

The later examples of urban flooding specifically relate to the local hourly rainfall extremes considered here, but none of these have been formally linked to climate change.

Does the aggregation to the whole of the UK (e.g. Fig S10) make sense? The records will be dominated by the regions with highest precipitation. Would it make sense to normalise the data? I suggest to add a short discussion.

We have looked at both the exceedance of UK records (orig Supp Fig S10, now Supp Fig S13) and regional records (Supp Fig S14). We feel both are important. UK records correspond to events that are of national importance and often reported in the media. Although they may be dominated by specific regions with the highest precipitation, we note that for hourly precipitation extremes return levels are much more uniform across the UK than is the case for daily precipitation extremes. Regional records correspond to locally unprecedented events, which may be of more interest to most stakeholders.

We have added a short note to justify looking at both UK and regional records.

Overall assessment

In its current version, I believe the manuscript is too unfocussed for a letter format, and not novel enough to merit publication in a high impact journal such as Nature Communications. I would rather suggest to expand the intercomparison with the standard RCM, to put the paper into the broader context of important scientific developments in the fields of signal emergence, trends of extreme (also hourly) precipitation and convection permitting simulations, and then to submit a considerably longer manuscript to a disciplinary journal such as Climate Dynamics or maybe Communications Earth & Environment.

As outlined above we have expanded the intercomparison with the RCM, and provided more background on the novelty of this study compared to other important developments in convection-permitting modelling. We also briefly introduce the concept of 'emergence', citing relevant literature, and point out the relevance of our analysis to this field of research. With these changes, the manuscript still meets the length requirements for Nature Communications.

Reviewer #3 comments

The authors present a comprehensive assessment of extreme rainfall events over the United Kingdom using an ensemble of high-resolution simulations in historic and future climates. They show trends in the magnitude and frequency of heavy rainfall events and quantify the uncertainty due to internal variability. They also show how the magnitude of heavy rainfall events scale with local temperature increases and find that the Clausius-Clapeyron relationship is not always followed and can

vary in different locations and different time periods. They also show that there are temporal variations in the number of records broken over time, with large gaps in record-breaking events, however, with a higher number of records still being broken due to climate change enhancement.

Overall, this paper makes a useful contribution to the climate literature. It makes use of a rare type of climate dataset: a high-resolution, convection-permitting, almost century-long medium-sized ensemble. This type of dataset, though computationally intensive, can uncover the role of climate change and climate variability on change in rainfall on very local scales, and can better capture the processes that lead to these types of events. A coarser model or a smaller ensemble would not be able to do that. The analysis that the authors carry out is sound and interesting.

Thank you for your positive and constructive review. Your comments have been very helpful in improving the paper.

However, there are points where the authors did not take the opportunity to really let this dataset and analysis shine (see comments about Figures 3 and 5). Additionally, there are areas in the text where the authors make conclusions that are not well-backed, and where methods are unclear. Therefore, I recommend that the paper be returned for revisions. Below, I outline my comments and suggestions.

We have taken the opportunity that Nature Comms affords to provide a more comprehensive manuscript, compared to Nature Climate Change, and have added text and additional supplementary figures addressing all your suggestions. Please see detail below.

Lines 68-70: Please give brief descriptions of the CPM and RCM model and simulation set ups here. I know they are in the methods, but I think this is important context for the results you are presenting here.

A brief description has been added to the start of the Results section, as suggested. Further detail is provided in the Methods. Some additional background material on CPMs has also been added to the Introduction.

Line 71: Please give a brief description of this dataset.

The CEHGEAR observations are already described in the Methods, but a brief description has been added in the Results section at first mention. Further detail on the reliability of the observational values, in particular over northern Scotland where the hourly gauge density is low, has also been added to the Results.

Line 73: In the methods, the authors state that some large rainfall events may not be represented in the observational dataset. I am curious to know if these omissions would be close to some of the values that the RCM seems to be overestimating?

As stated in the Methods hourly gauges are expected to underestimate the intensity of heavy rainfall events due to under-sampling and under-catch, however the disaggregation processes whereby “an average storm profile is used to disaggregate the daily dataset, where the nearest gauge was >50km away” will also lead to additional error. In particular if rainfall is more persistent in time than the average UK event, as is the case over orography, then the disaggregation step could actually lead to an overestimation of peak values. This is now briefly discussed in the Results, in the context of model biases over northern Scotland.

In the case of the values exceeding 100mm/h at the 12km scale in the RCM, these events are unphysical and the under-sampling of heavy events in the gauge dataset is not expected to explain such high values. The maximum hourly value in the CEHGEAR observations for rainfall regridded to the 12km scale is 63mm/h, and systematic gauge under-catch is expected to give biases of on average 20% (although biases can be as large as 50% in winter or over high terrain, Rajczak & Schär, 2017). Even allowing for a bias of 20%, this does not give values above 100mm/h.

A note to this effect has been added to the Methods, and the following reference has been added:

Rajczak, J., & Schär, C. (2017). Projections of future precipitation extremes over Europe: A multi-model assessment of climate simulations. *Journal of Geophysical Research: Atmospheres*, 122, 10, 773–10,800. <https://doi.org/10.1002/2017JD027176>

Line 88-90: I think this sentence warrants some elaboration especially for a journal with a broader audience. Could you please explain "unphysical grid point storms" a little further?

An explanation is now provided in the text:

“Such an event is not physical but instead is a numerical instability in the model, and arises due to violation of the assumption that convective clouds have areas much smaller than the grid-box.”

Line 90: How do you know that these values are "unphysical" and not just due to the lack of representation of internal variability in the observations? Please elaborate.

We now make the point in the text that these values greater than 100mm/h at the 12km scale are considerably higher than observed maximum values even accounting for gauge under-catch, with additional detail added to the Methods (see response to comment above).

We note that high hourly values exceeding 100mm/h at the local station (or point) scale do occur in the UK (e.g. Burt 2005, Golding et al 2005), but such spatial scales are much smaller than the RCM is able to resolve.

References:

Burt, S., (2005): Cloudburst upon Hendrabortnick Down: The Boscastle storm of 16 August 2004. *Weather*, 60 (8), 219–227, doi:10.1256/wea.26.05.

Golding, B., P. Clark, and B. May, (2005) The Boscastle flood: Meteorological analysis of the conditions leading to flooding on 16 August 2004. *Weather*, 60 (8), 230–235, doi:10.1256/wea.71.05.

Figure 1 panel d. Please indicate which box plots correspond to which label on the y-axis. From the

current labels, it is not clear.

This is now clarified in the figure caption.

Line 113: Please note here that this is a visual inspection and not a calculated trend.

Done.

Line 138: Could you please define "lots" of extreme rainfall events in a more specific and quantitative way? It seems a bit hand-wavy right now.

The following sentence gives an example of lots of extreme events (e.g. 70 in one year) and an example of few events (e.g. 10 in one year). Beyond this, we do not think it appropriate to give a formal definition.

Line 141: Typo - m/h  mm/h

Change made.

Figure 3: Why do the authors not show grid-point values of these values and ratios on three maps, with each grid points shading representing the ensemble-average (1) historic and (2) future values, and the (3) ratio? Given the power of the CPM simulations to allow for hyper-local information, I think the authors are missing a big opportunity to visually show the importance of their fine scale runs and the localization of extreme rainfall events.

Figure 3 is now Fig 4 in the revised manuscript. We have presented changes in a regional-aggregated sense as this allows us to provide more robust estimates of change, whereby all exceedances of the threshold locally at the grid point scale within the region are counted.

We have tried doing the analysis for individual grid-points, with values displayed on a map as suggested (Fig R6). It can be seen that the ratio plot is noisy, generally reflecting grid-point scale differences in small numbers of events in the 1980s (with many grid boxes in Scotland having no events at all). From the maps it can be seen that there are large increases (greater than 10x) in the numbers of events locally over many grid points in Scotland and over some (but fewer) grid points in England and Wales, with more grid points showing smaller increases (3x or less) in the south. Thus the general picture emerging from the regional-aggregated maps (Fig 4) is also seen at the local scale. The maps showing the grid-scale information in the numbers of events in the 1980s and 2070s have been added to the Supplementary material (Supp Fig S8), but the ratio plot is not shown as the local differences in changes are unlikely to be robust. This is now discussed in the manuscript.

Fig R6 (first panels are Supp Fig S8): Average number of events per year exceeding 20mm/h locally in the 1980s and 2070s, and their ratio. Values correspond to the multi-member 10-year mean number of events, for hourly rainfall averaged over 12km grid box in the CPM. Grey indicates grid boxes outside the UK or where the ratio values cannot be calculated due to there being no events in the 1980s.

Figure 3: Some of the text on the map is difficult to read. Suggestion to change colors to improve readability, or to separate the bar plots from the map.

The colours in both Figures 4 and 6 (old Figs 3 and 5) have been modified (made more transparent) and the text has been made bold to improve readability.

Line 170: Please specify that this is degree Kelvin.

Done

Lines 199-202: Could you please elaborate on this conclusion? I'm not sure I am able to easily connect the dots between the scatter and the importance of sub-seasonal variability.

Text has been added to elaborate on this as follows: "The temperature at the time of extreme precipitation events is a more direct measure of moisture availability, and this will depart considerably from the annual mean temperature due to sub-seasonal variability. In addition dynamical drivers, related to regional circulation patterns, will be important in determining the year-to-year variability in the intensity and occurrence of precipitation extremes."

Figure 5: Like my comment about Figure 3, (1) readability of some the text needs to be improved, and (2) can the authors show the grid-point level data for the scaling coefficients?

As for Fig 4 (old Fig 3), the colours in Fig 6 (old Fig 5) have been made more transparent and the text has been made bold to improve readability.

A map showing grid-point level data for the scaling coefficients has also been added as a new figure to the paper (Fig 7, copied below). Thank you for this suggestion, as we agree it highlights the ability of the new CPM ensemble to robustly represent changes in local rainfall extremes. We note that the scaling coefficient is less noisy than the map showing the changes in the number of events per year (Fig R6 above) since the full 100y timeseries of annual maximum data at each grid box is used to fit a linear regression line to estimate the scaling, whereas for the events map only values exceeding a very high threshold in two 10-year periods are used.

The local maps confirm the picture of a north-south gradient in scaling coefficients, with grid-box values close to or below CC scaling in the south and east. Higher scaling coefficients are seen in the north of the UK and over western coasts, with values above 2xCC locally in the north-west. This discussion has been added to the paper.

Fig 7: Scaling coefficients (%/K) for underlying climate change signal in local annual maximum hourly precipitation in the CPM. Values correspond to the gradient of a linear trend line fitted to smoothed

annual maximum precipitation versus local annual mean temperature at the 12km scale, expressed as % increase in precipitation per K temperature increase. The smoothed data consists of the multi-member 10-year running mean.

Line 267: Could you please elaborate on why this is not realistic? Also, please clarify what you mean by "grid point storms".

It is not the jump itself that is unrealistic but rather the very high (>100mm/h at 12km scale) record value. This is now clarified in the text.

Additional text briefly explaining what a grid point storm is has been added to the validation section of the Results (see response to comment above).

Line 311: Typo: no evidence  not evidence

Change made

Line 317: Please specify which high emission scenario (i.e., RCP8.5).

Done

Line 365: Please clarify that you are using twelve simulations from the SAME RCM and not simulations from twelve DIFFERENT RCMs.

Note that this is a perturbed parameter ensemble and thus the different RCM ensemble members are different versions of the same RCM. This has been clarified in the text.

In the following paragraph more detail is provided on the different RCM ensemble members. In particular, these are created by perturbing uncertain parameters in the RCM model physics. This is different to the CPM ensemble where no parameter perturbations are applied, and thus the same CPM is used for each member. Perturbations were not applied to the CPM members, since given the different model physics, it was not possible to mirror the perturbations applied in the driving RCM members. This is explained in the following paragraph.

Lines 378-381: I am not sure that your results showing the ensemble spread ONLY reflect internal variability as stated in the main text but may also reflect model physics uncertainty. This is because I am assuming the boundary conditions for each ensemble will be different because of the different physics parameterizations of the driving model. In that case, I recommend additional explanation or reasoning in terms of what is reflected in the spread of the ensemble.

As clearly stated in the "Models and observational data" section of the Methods, "the CPM ensemble samples uncertainty arising from natural variability and uncertainty in the physics of the driving models" i.e. ensemble spread does not only reflect internal variability. However we have added a note that internal variability dominates CPM ensemble spread for future changes in hourly precipitation extremes, whilst model uncertainty dominates for the RCM (Fosser et al, 2020).

The following reference has been added:

Fosser, G., E. J. Kendon, D. Stephenson, S. Tucker (2020) Convection-Permitting Models Offer Promise of More Certain Extreme Rainfall Projections, *Geophys. Res. Lett.* 47 (13), <https://doi.org/10.1029/2020GL088151>

In the analysis of records the influence of natural variability is determined by analysing the occurrence of records in detrended data. This is data for each member where the underlying climate change signal (given by the running 10-year multi-member mean) has been removed. This is described in the Methods. We note that taking the multi-member mean has the advantage of reducing any influence from variability on multi-decadal timescales, since different ensemble members are not in phase in their different realisations of variability. However, it does also average across the effects of the different parameter perturbations in the driving model. Since parameter perturbations in the RCM have a relatively weak influence on future changes in the CPM (Fosser et al 2020), this is valid, with the underlying climate change signal expected to be the same across CPM members. This is now discussed in the Methods.

REVIEWERS' COMMENTS

Reviewer #2 (Remarks to the Author):

Review of revised manuscript "Variability conceals emerging trend in 100yr projections of local hourly rainfall extremes" by Kendon et al.

The authors argued convincingly about key novel aspects in their manuscript and have greatly improved the overall text.

I would, however, recommend to work over the abstract. The discussion section raises key points about the relevance of the results, but the abstract focuses on numbers which to a large extent are in principle known already from existing time slices experiments. The final sentence in the abstract is just a common place which could easily be deleted to provide more relevant information. The focus there should be on the relevance of internal variability and the clustering of records.

Reviewer #3 (Remarks to the Author):

The authors have successfully revised the manuscript and have responded to my comments sufficiently. I believe that the paper provides important insight into extreme precipitation and the need to (1) resolve convection in climate models for more representative precipitation values, and (2) capture internal variability sufficiently to quantify changes in the statistics of extreme precipitation.

I have some minor comments that the authors should address before the paper is ready to be accepted:

Title: I think an indication that this is a U.K.-specific study might be needed here - but I will leave it up to the editor and will note that to them.

Line 54: Please expand "UKCP" when introducing it for the first time.

Line 79: no hyphenation needed for "first time".

Figure 2: Please include panel labels (a, b, c...) in figure and caption

Lines 208-210: Typically, one would use a spatially varying percentile threshold instead of a spatially common one to understand the locally-relevant changes in extreme precipitation. I'm not sure the percentile analysis really adds value here since the spatially common threshold is very close to the 10mm/hour threshold already analyzed.

Figures 4 and 6: Please use a more descriptive figure title rather than just "UK".

Line 258: "th" typo.

Lines 267-269: I am supportive of using local mean temperatures for understanding the role of C-C scaling in changes in precipitation, and do not suggest changing your analysis at all. However, I am curious if using local temperatures for scaling, and not global or regional, would miss some of the changes in precipitation due to changes in weather patterns or even more large-scale atmospheric processes. I would think that local temperatures would not necessarily reflect the influence of these non-local and more regional processes. I think a brief discussion on this would be interesting and useful here.

Lines 276-277: This sentence is not very clear to me - do you mean that RCMs would not show similar variability in the scaling rates because they do not resolve the dynamical processes that are resolved in the CPM? Please clarify.

Figure 5 labeling and referencing: Please include panel labels (a, b, c, etc) to avoid positional descriptions of the panels (top, bottom, etc).

Figures 6 and 7: Optional suggestion to combine figures 6 and 7 into one figure with two panels since they are showing the same thing but figure 6 is regionally averaged and seasonally separated.

Variability conceals emerging trend in 100yr projections of UK local hourly rainfall extremes

By Kendon EJ, EM Fischer and C Short

Nature Communications manuscript NCOMMS-22-22649-A

Reviewer response document

Please find below detailed responses to the reviewers' comments, with responses in red text. Also uploaded is a manuscript file with revisions marked as tracked changes.

Reviewer #1 comments

The authors have addressed my questions. And I appreciate the authors' response to all the comments of the reviewers. I recommend accepting this paper.

We thank the reviewer for their time in reviewing our paper.

Reviewer #2 comments

The authors argued convincingly about key novel aspects in their manuscript and have greatly improved the overall text.

We thank the reviewer for their helpful comments.

I would, however, recommend to work over the abstract. The discussion section raises key points about the relevance of the results, but the abstract focuses on numbers which to a large extent are in principle known already from existing time slices experiments. The final sentence in the abstract is just a common place which could easily be deleted to provide more relevant information. The focus there should be on the relevance of internal variability and the clustering of records.

We have changed the last sentence of the abstract to focus on the novel information around clustering of extreme events and its implications. Other minor changes have also been made to the abstract to improve readability. The revised abstract is as follows:

“Extreme precipitation is projected to intensify with warming, but how this will manifest locally through time is uncertain. We exploit the first ensemble of convection-permitting transient simulations to examine the emerging signal in local hourly rainfall extremes over 100-years. We show rainfall events in the UK exceeding 20mm/h that can cause flash floods are 4-times as frequent by 2070s under high emissions; in contrast, a 12km regional model shows only a 2.6x increase. With every degree of regional warming, the intensity of extreme downpours increases by 5-15%. Regional records of local hourly rainfall occur 40% more often than in the absence of warming. However, these changes are not realised as a smooth trend. Instead, as a result of internal variability, extreme years with record-breaking events may be followed by multiple decades with no new local rainfall records. The tendency for extreme years to cluster poses awkward challenges for communities trying to adapt.”

Reviewer #3 comments

The authors have successfully revised the manuscript and have responded to my comments sufficiently. I believe that the paper provides important insight into extreme precipitation and the need to (1) resolve convection in climate models for more representative precipitation values, and (2) capture internal variability sufficiently to quantify changes in the statistics of extreme precipitation.

We thank the reviewer for their very constructive review, and have addressed all the additional minor comments they raise below.

I have some minor comments that the authors should address before the paper is ready to be accepted:

Title: I think an indication that this is a U.K.-specific study might be needed here - but I will leave it up to the editor and will note that to them.

We have added UK to the title as suggested.

The fact that this is a UK study is also stated in the abstract. We note that the key findings of this paper relating to (1) the importance of resolving convection for projecting changes to local precipitation extremes; (2) the role of internal variability in record breaking behaviour and (3) the finding that extreme rainfall changes are not realised as a smooth trend, with periods of rapid intensification following by pauses, apply more generally and not just to the UK.

Line 54: Please expand "UKCP" when introducing it for the first time.

Done

Line 79: no hyphenation needed for "first time".

This use of "first time" has now been deleted in response to the editor's comments.

Figure 2: Please include panel labels (a, b, c...) in figure and caption

Done

Lines 208-210: Typically, one would use a spatially varying percentile threshold instead of a spatially common one to understand the locally-relevant changes in extreme precipitation. I'm not sure the percentile analysis really adds value here since the spatially common threshold is very close to the 10mm/hour threshold already analyzed.

The value in using the percentile threshold here is in the comparison of models. The percentile (99.99th) was deliberately chosen to be close to the 10mm/h threshold, so that we are considering equivalently extreme events to the absolute threshold results in Supp Table S1. The percentile threshold analysis allows us to confirm that the differences between the CPM and RCM, in terms of future changes in the number of events exceeding a high threshold, is not simply a reflection of the

absolute value being less extreme in the RCM. Using the percentile threshold is essentially a simple form of bias adjustment. Even if we do this adjustment there is still a significantly greater future increase in the number of events in the CPM than RCM. Thereby, we demonstrate that the greater increase is not just an artefact of different climatologies. This emphasizes the importance of resolving convection for projecting future changes in precipitation extremes.

We therefore prefer to keep the percentile analysis.

Figures 4 and 6: Please use a more descriptive figure title rather than just "UK".

The "UK" was referring to the subpanel with results for the UK region and was not intended as a figure title. Since this is described in the figure caption, we have just removed this subpanel title.

Line 258: "th" typo.

Change made

Lines 267-269: I am supportive of using local mean temperatures for understanding the role of C-C scaling in changes in precipitation, and do not suggest changing your analysis at all. However, I am curious if using local temperatures for scaling, and not global or regional, would miss some of the changes in precipitation due to changes in weather patterns or even more large-scale atmospheric processes. I would think that local temperatures would not necessarily reflect the influence of these non-local and more regional processes. I think a brief discussion on this would be interesting and useful here.

That's a good point. Following the reviewer's suggestion, we include a brief discussion on this (see below).

In Fig 7 we use local mean temperatures for scaling, indicative of changes in moisture availability locally (under the assumption of constant relative humidity). However, we have also tried using UK-average temperature, rather than the local grid box value to calculate scaling (added as new Supplementary Fig S10, copied below). This gives some indication of the extent to which results may be different on using a more regional temperature measure, that is indicative of changes in moisture availability in the wider region. It can be seen that the scaling results are very similar when using local or UK-mean temperature. There are fewer locations showing low scaling coefficients ($<5\%/K$) in the south and fewer showing very high scaling coefficients ($>12\%/K$) in the north, on using the more regional temperature measure. This suggests that some of the low and very high scaling rates, that depart significantly from CC-scaling, may indeed be due to the large-scale moisture availability in weather systems (where the moisture source may be remote from the local region) increasing at a different rate to the local temperature-dependent maximum. Whether local or regional temperature is the appropriate scaling variable will depend on the relative importance of large-scale weather patterns as opposed to locally triggered convection in generating precipitation extremes locally, which will vary by region and season. Nevertheless, the fact that results are very similar on using local or UK-mean temperature suggests that the conclusions here are not sensitive to this choice. In particular, we still find evidence that values are close to CC-scaling for southern/central regions, with higher values (up to $2\times CC$ locally) in the north-west of the UK.

We have added a brief discussion of this to the text, along with the new supplementary figure:

“We note that results are very similar on using UK-mean instead of local temperature as the scaling variable (Supp Fig S10), although the tendency for large departures from CC-scaling is reduced. This suggests that some of the low (in the south) and high (in the north) scaling rates (Fig 7) may be due to moisture availability in weather systems (where the moisture source may be remote) increasing at a different rate than the local temperature-dependent maximum. The appropriate scaling variable will depend on the relative importance of large-scale weather patterns compared to locally triggered convection. However, the similarity of the results suggests that our conclusions here are not sensitive to this choice.”

Supp Fig S10: Scaling coefficients (%/K) for underlying climate change signal in local annual maximum hourly precipitation in the convection-permitting model (CPM). Values correspond to the gradient of a linear trend line fitted to smoothed annual maximum precipitation versus annual mean temperature, (left) using local temperature at the 12km scale and (right) using UK-average temperature, expressed as % increase in precipitation per K temperature increase. The smoothed data consists of the multi-member 10-year running mean.

Lines 276-277: This sentence is not very clear to me - do you mean that RCMs would not show similar variability in the scaling rates because they do not resolve the dynamical processes that are resolved in the CPM? Please clarify.

Because RCMs do not capture local dynamical processes, we do not expect them to show the same seasonal and regional variation in scaling rates. This is because local dynamical feedbacks are expected to be particularly important for short-duration convective storms, that predominantly occur in summer. Looking at Supp table S2, we see that scaling coefficients for the raw data are

consistently higher in the CPM than RCM for all but one region in summer; whereas they are typically lower in the CPM in winter. However, the relationship between CPM and RCM scaling rates is not straightforward, and there are many contributing factors (some of which are discussed in the text relating to departures from CC-scaling) – so it is not possible to provide a simple physical explanation. This is also made more difficult by unphysical grid point storms having a significant influence on simulated extreme precipitation values in the RCM.

We have modified the text slightly to try and make this clearer:

“Such local dynamical feedbacks are only captured by CPMs, and thus we may anticipate different scaling rates between the CPM and RCM. For the raw data, scaling coefficients are higher in the CPM than RCM for summer extremes (consistent with the importance of local dynamical feedbacks within convective storms) and typically lower for winter extremes; but the pattern is not straightforward for the underlying climate change signal (Supp Table S2). This is due to differences in scaling reflecting many factors, including the presence of grid point storms in the RCM.”

Figure 5 labeling and referencing: Please include panel labels (a ,b, c, etc) to avoid positional descriptions of the panels (top, bottom, etc).

Change made

Figures 6 and 7: Optional suggestion to combine figures 6 and 7 into one figure with two panels since they are showing the same thing but figure 6 is regionally averaged and seasonally separated.

We are happy to make this change if the reviewer/editor feels this would be better. Our preference, however, is to keep them separate, as this allows the sizing of the figures to be different. In particular, it is beneficial for Fig 6 (and equivalently Fig 4) to be large, so that the inset bar charts are easy to read. Figure 7 by comparison can be smaller.